# Acetylcholine receptor based chemogenetics engineered for neuronal inhibition and seizure control assessed in mice

Quynh-Anh Nguyen [1,4] ✉, Peter M. Klein [1,4] ✉, Cheng Xie[2], Katelyn N. Benthall[2], Jillian Iafrati[2], Jesslyn Homidan[1], Jacob T. Bendor[2], Barna Dudok [1,3], Jordan S. Farrell [1], Tilo Gschwind[1], Charlotte L. Porter [1], Annahita Keravala[2], G. Steven Dodson[2] & Ivan Soltesz [1]

Epilepsy is a prevalent disorder involving neuronal network hyperexcitability, yet existing therapeutic strategies often fail to provide optimal patient outcomes. Chemogenetic approaches, where exogenous receptors are expressed in defined brain areas and specifically activated by selective agonists, are appealing methods to constrain overactive neuronal activity. We developed BARNI (Bradanicline- and Acetylcholine-activated Receptor for Neuronal Inhibition), an engineered channel comprised of the α7 nicotinic acetylcholine receptor ligand-binding domain coupled to an α1 glycine receptor anion pore domain. Here we demonstrate that BARNI activation by the clinical stage α7 nicotinic acetylcholine receptor-selective agonist bradanicline effectively suppressed targeted neuronal activity, and controlled both acute and chronic seizures in male mice. Our results provide evidence for the use of an inhibitory acetylcholine-based engineered channel activatable by both exogenous and endogenous agonists as a potential therapeutic approach to treating epilepsy.

Around 50 million people worldwide have epilepsy, making it one of the most prevalent neurological disorders[1]. The traditional standard of care is treatment with systemic anti-seizure medications, which alter brain-wide activity and often carry significant side effects[2–4]. Indeed, while effective seizure control remains the top concern for people with epilepsy, patients and caregivers additionally place nearly equal importance on treatments that minimize adverse side effects and best address quality of life issues[5]. About one-third of epilepsy patients will not have their seizures fully controlled by anti-seizure medications, a proportion that has little changed over the last 100 years, despite the accelerated pace of drug development in the last few decades[6–8]. Therefore, patients must often turn to more invasive treatments, such as surgical resection or ablation of the underlying epileptic tissue, which can carry a range of significant irreversible effects[9]. Devising effective therapeutic strategies to specifically modulate neurons

underlying focal seizure generation has the potential to more precisely constrain epileptiform activity while sparing other critical brain functions.

Using genetic approaches to target the expression of proteins capable of constraining neuronal excitability into select epileptogenic network elements is an intriguing alternative therapeutic strategy. Introducing additional potassium channels[10–12] or hyperexcitability-activated chloride channels[13] in order to hyperpolarize neurons can effectively suppress seizures in animal models of epilepsy. However, such gene therapy approaches lack external regulatory mechanisms that allow for ongoing titration of neuronal suppression to levels that sufficiently control seizures while not producing excessive side-effects. Optogenetic strategies enable highly temporally precise control of activity in select neurons, allowing regulated, closed-loop, termination of seizures[14–17]. While these approaches show promise in

[1]Department of Neurosurgery, Stanford University, Stanford, CA 94305, USA. [2]CODA Biotheapeutics, 240 East Grand Ave., South San Francisco, CA 94080, USA. [3]Department of Neurology, Baylor College of Medicine, Houston, TX 77030, USA. [4]These authors contributed equally: Quynh-Anh Nguyen, Peter M. Klein. ✉e-mail: qanguyen@stanford.edu; kleinp@stanford.edu

treating seizures at a preclinical level, concerns related to the need for invasive methods to deliver light in deep brain structures, light-associated tissue heating, and introducing non-mammalian proteins into the brain remain hurdles to clinical translatability[18–20].

Chemogenetic approaches avoid some of these potential limitations, providing targeted control of neurons exogenously expressing modified mammalian receptor proteins that are engineered for selective activation by systemically administered agonists[21–25]. The Designer Receptors Exclusively Activated by Designer Drugs (DREADD) technique has been found to effectively control seizures in some preclinical animal models of epilepsy[26–28]. However, the use of the inhibitory Gi-coupled hM4Di DREADD system relies on downstream effectors, such as G protein-activated inwardly rectifying potassium (GIRK) channels that can be disrupted in epilepsy[29,30]. Activation of Gi signaling can also modulate non-specific second messenger pathways[24,31]. In addition, the most commonly used DREADD agonist, clozapine-N-oxide (CNO), can also be metabolically converted to clozapine, which binds to some off-target receptors in the brain[32–34].

An alternative chemogenetic approach uses chimeric engineered ligand-gated ion channels (eLGICs) to enable more direct control of ion fluxes. Of these, a class of channels has been developed based on fusing the ligand-binding domain of the α7 nicotinic acetylcholine receptor (nAChR) to one of various ion pore domains. An advantage of these channels is that α7 nAChRs are comprised of homopentameric subunits, requiring expression of only a single subunit for functional assembly[35]. In addition, variants of these channels, such as the pharmacologically selective actuator modules (PSAMs), can be engineered to selectively enhance responses to known clinically approved agonists or other ligands[25,36]. However, the potential application of these channels for the treatment of epilepsy has not been previously investigated.

Here we show that our newly developed eLGIC named Bradanicline- and Acetylcholine-activated Receptor for Neuronal Inhibition (BARNI), containing a modified ligand-binding domain of the α7 nAChR coupled to the anion-permeable pore domain of the α1 glycine receptor, is able to effectively inhibit neuronal activity. We use the α7 nAChR-selective agonist bradanicline (TC-5619), which is an appealing drug candidate that has already been evaluated through multiple phase 2 clinical trials and been shown to be safe and well tolerated[37,38]. Activation of BARNI in vivo led to an increase in the threshold for electrically evoked seizures in wild-type mice and a decrease in the frequency of spontaneous seizures in chronically epileptic mice. We utilize a recently developed genetically encoded fluorescent ACh sensor to show that seizures result in large increases of ACh in the brain, suggesting channels with maintained endogenous ACh ligand-binding could have an on-demand effect in controlling seizures.

## Results

### The eLGIC BARNI enables regulation of hippocampal neuronal excitability

Our experiments evaluated the newly generated BARNI channel, which is a chimeric ligand-gated ion channel comprised of a modified α7 nAChR ligand-binding domain fused to the chloride-permeable ion pore domain of the α1 glycine receptor (Fig. 1a, b). Distinct from the PSAM eLGICs[25,36] the BARNI extracellular domain contains the Cys-loop of the glycine receptor (Supplementary Fig. 1), which is required for fast gating of the chimeric receptor[39]. As opposed to other similar eLGICs[25,36,39], the BARNI channel utilizes the α1Ins splice variant of the α1GlyR, which includes an extended M3–M4 loop portion of the ion pore domain that may enhance peak currents[40]. We tested in cultured mouse hippocampal neurons the effect of receptor activation by the drug bradanicline, which is a highly selective α7 nAChR agonist[41]. The BARNI channel shows much greater sensitivity to bradanicline ($EC_{50} = 0.019\,\mu M$) than for ACh ($EC_{50} = 95.1\,\mu M$) (Supplementary

Fig. 2a). In dissociated hippocampal neurons expressing BARNI we found a dose-dependent decrease in neuron firing frequency with increasing bradanicline concentrations (Supplementary Fig. 2b). Unlike the wild type α7 nAChR, which displays rapid desensitization when activated[42], decreased excitability in BARNI-expressing dissociated neurons was sustained across 20 min bradanicline exposures (Supplementary Fig. 2c–f).

To determine whether activation of BARNI by bradanicline could control neuronal activity ex vivo, we performed whole-cell patch clamp recordings in acute brain slices from mice virally transfected with vectors containing a pan-neuronal hSyn-BARNI expression cassette or control scrambled cassette into the hippocampus. Neurons in the CA1 subregion were selected for recording based on co-expression of GFP from the same vector, and action potential dynamics were monitored before, during, and after a 1-min 0.15 μM bradanicline bath application. We used a stepwise current injection followed by a continuous ramp protocol to monitor input resistance, rheobase, and action potential count throughout each recording (Fig. 1c). Bradanicline exposure produced a clear decrease in the input resistance of BARNI-expressing neurons, restricted to the period following drug application (Fig. 1d, e, Supplementary Fig. 3), consistent with effective channel opening. BARNI-expressing neurons also underwent increases in the rheobase current needed to evoke action potentials (Fig. 1f, g) and a decrease in the number of action potentials generated in response to depolarization (Fig. 1h), which were again most altered in the period after bradanicline exposure. These results support the ability of BARNI to rapidly and robustly decrease the excitability of hippocampal neurons when expressed and activated in the mouse brain. We also observed similar lasting suppression in an independent cohort of animals (Supplementary Fig. 4), demonstrating the reproducibility of our findings. BARNI channel activation was comparably effective in decreasing neuronal excitability within both putative pyramidal cell and interneuron subsets of the recorded neurons (Supplementary Fig. 5). Our electrophysiology results are due to direct effects of bradanicline on BARNI, since they do not occur in scramble-expressing neurons and are occluded by the α7 nAChR antagonist α-bungarotoxin (Supplementary Fig. 6).

### Increased threshold for electrically evoked seizures by BARNI activation

The dramatic suppression of neuronal excitability we observed in hippocampal slices led us to examine whether this chemogenetic approach could be used in vivo to treat epilepsy, which is characterized by aberrant excessive neuronal firing leading to the emergence of seizures. We first evaluated whether hippocampal BARNI expression, paired with bradanicline administration, was able to modify the threshold for focal seizures evoked through stimulation of the perforant path projections from the entorhinal cortex to the hippocampus. We performed bilateral injections of an AAV vector containing an hSyn-driven BARNI expression cassette or control scrambled cassette into the hippocampus, paired with implantation of a stimulating electrode in the perforant path and a recording electrode in the ipsilateral dorsal hippocampus. We administered intraperitoneal (i.p.) injections of either bradanicline (100 mg/kg) or saline vehicle buffer prior to subjecting mice to increasing electrical stimulation intensities until a sufficient threshold for evoking a seizure was observed. At our selected dosage of peripheral administration, we found that bradanicline concentrations measured in plasma and brain tissue remain above the $EC_{50}$ for BARNI channel activation in neurons for several hours, demonstrating effective penetrance of the compound across the blood-brain barrier (Supplementary Fig. 7).

We were able to reliably evoke seizures detectable in the CA1 region of the hippocampus following supra-threshold electrical stimulation of perforant path inputs (Fig. 2a). Across all vector and drug treatment groups, we observed a range of seizure threshold potentials

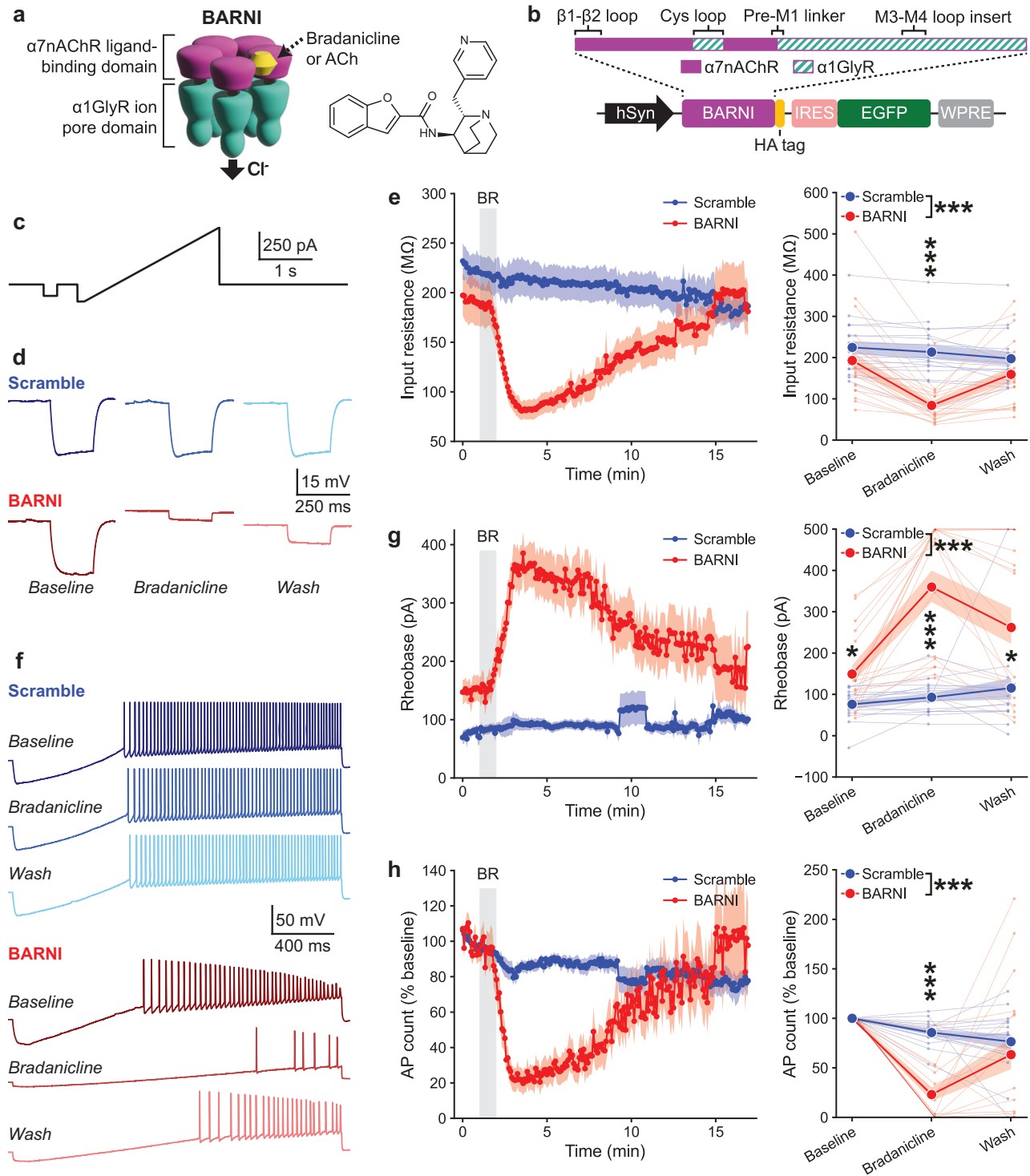

from 30 to 180 µA. Notably, we observed that only BARNI-expressing mice were more resistant to having hippocampal seizures evoked in trials where they received bradanicline, due to elevated thresholds (Fig. 2b, c). We observed no impact of vector injection or bradanicline administration on evoked seizure durations (Fig. 2d, e). In a separate cohort of freely exploring mice where we performed CA1 local field potential (LFP) recordings, we observed no baseline impacts of BARNI expression on cognition-related hippocampal LFP oscillations, or any within animal shifts in LFP properties for our groups following bradanicline injection (Supplementary Fig. 8). Building upon the success of our chemogenetic approach in reducing susceptibility to evoked seizures, we next wanted to examine if our BARNI eLGIC could be

effective in a chronic epilepsy model, which displays seizures that more closely resembles those in humans.

## Amelioration of spontaneous seizures in a mouse model of chronic focal TLE

To test whether BARNI expression and bradanicline administration could also control chronic spontaneous seizures, we utilized the intrahippocampal kainic acid (IHKA) mouse model of temporal lobe epilepsy (TLE)[43–45]. The IHKA mouse model recapitulates several hallmarks of highly refractory temporal lobe epilepsy, including focal hippocampal sclerosis and seizures that mainly emerge near the sclerotic region. Mice received an initial unilateral injection of kainic

**Fig. 1 | BARNI channel activation suppresses neuronal excitability. a** Schematic of the chimeric BARNI chemogenetic receptor and bradanicline (TC-5619) chemical structure. **b** BARNI channel expression construct. hSyn human synapsin promoter, IRES internal ribosome entry site, WPRE posttranscriptional regulatory element. **c** Current clamp protocol for monitoring input resistance and action potential dynamics of transduced CA1 neurons in acute hippocampal slices. **d** Representative traces of voltage responses to −100 pA stepwise current injections before, during, and after washout of bradanicline (0.15 μM) in BARNI- and Scramble-expressing cells. **e** *Left*, time course of input resistance changes in response to 1 min bath application of bradanicline (BR), sampled every 5 s. *Right*, Time-binned input resistance values for individual animals (small circles) and within groups (large circles) show a greater decrease in BARNI vs. Scramble neurons ($F_{(1, 100)} = 20.45$, $P = 1.7e{-}5$), occurring specifically after bradanicline application ($F_{(2, 10)} = 4.59$, Baseline: $P = 0.820$, Bradanicline: $P = 5.0e{-}6$, Wash: $P = 0.454$). **f** Representative traces of action potentials observed as increasing current are injected in the ramp protocol before, during, and after washout of bradanicline (0.15 μM) in BARNI- and

Scramble-expressing cells. **g** *Left*, Time course of changing rheobase currents required to evoke action potentials in response to 1 min bath application of bradanicline. *Right*, Time-binned rheobase values show a rise in BARNI vs. Scramble neurons ($F_{(1, 98)} = 45.22$, $P = 1.2e{-}9$), that increased after bradanicline application and persisted into the wash phase ($F_{(2, 98)} = 5.52$, Baseline: $P = 0.013$, Bradanicline: $P = 3.0e{-}6$, Wash: $P = 0.027$). **h** *Left*, Time course of changes in action potential (AP) counts, normalized to the pre-drug average for each neuron, in response to 1 min bath application of bradanicline. *Right*, Time-binned action potential counts are reduced in BARNI vs. Scramble neurons ($F_{(1, 99)} = 13.97$, $P = 3.1e{-}4$), specifically after bradanicline application ($F_{(2, 99)} = 7.98$, Baseline: $P = 1.000$, Bradanicline: $P = 3.2e{-}8$, Wash: $P = 0.924$). Baseline values are averaged across 1 min prior to drug application, bradanicline values are 1–3 min after application and Wash values are >8 min after drug. Mean ± SEM, $n = 20$ cells/8 animals BARNI, $n = 16$ cell/9 animals Scramble. *$P < 0.05$, ***$P < 0.001$ (two-way ANOVA with two-sided Bonferroni multiple comparisons correction). Source data are provided as a Source Data file.

acid into the dorsal hippocampus to induce epilepsy (Fig. 3a). After allowing at least 3 weeks for the emergence of chronic spontaneous seizures, mice were then bilaterally injected with AAV vector encoding either an hSyn-BARNI expression cassette or a control scrambled cassette, and subsequently implanted with depth electrodes for 24-h EEG monitoring. After at least 3 further weeks for viral vector expression, mice were then subjected to i.p. injection of either bradanicline (100 mg/kg) or saline vehicle while monitoring seizure frequency and duration.

We observed clear reductions in spontaneous seizures after a single i.p. injection of bradanicline in mice expressing BARNI (Fig. 3b, d). Overall, we found a significant decrease in seizure frequency following administration of bradanicline in BARNI-expressing animals compared to either vehicle injection or Scramble-expressing animals, with these findings being consistent across both normalized and absolute seizure frequency quantifications and with the use of longer thresholds for seizure duration (Fig. 3c, Supplementary Figs. 9, 10). Notably, we determined that the observed decrease in seizure frequency was quite long-lasting, enduring up to 240–270 min after bradanicline administration, generally corresponding with the sustained presence of bradanicline in mouse brains for several hours after i.p. injection (Supplementary Fig. 7b). Indeed, when we administered a second dose of drug 2 h after the first, we were able to extend the time period of seizure frequency reduction to more than 10 h (Supplementary Fig. 11a) and this effect was consistent across several days (Supplementary Fig. 11b). Stronger inhibition of seizures may be associated with greater BARNI expression in the dentate gyrus than in the CA1 region of the hippocampus (Supplementary Fig. 12), suggesting that even more effective seizure control could be achieved by further optimizing BARNI localization. For the few remaining seizures that were observed after bradanicline administration in BARNI-expressing mice, we found a brief decrease in seizure duration immediately after bradanicline administration (Fig. 3e). However, such a measure does not account for the fully reduced duration of seizures that are successfully prevented. Overall, constraining hippocampal hyperexcitability with the BARNI eLGIC is an effective strategy to ameliorate spontaneous focal seizures in the etiologically relevant IHKA model of TLE.

### Seizures lead to increase of acetylcholine in the brain

Beyond the exciting therapeutic potential of chemogenetic approaches selectively activated by systemically administered compounds, there is also interest in whether engineered receptors can be tuned to control neuronal hyperexcitability in response to natural fluctuations of neurotransmitters during seizures[9,13]. When we analyzed the baseline seizure dynamics of our kainic acid injected animals, which were taken starting 1–2 days after virus injection, we noticed that the cumulative number of seizures in BARNI-expressing animals was

significantly reduced more than 3 weeks after virus injection (Supplementary Fig. 13), even in the absence of bradanicline. As previously mentioned, while the BARNI channel has high potency for bradanicline, it still maintains the ability to bind and be activated by endogenous ACh. Although steady-state concentrations of ACh (0.1–6 nM) are below what is needed to activate the BARNI receptor, elevated ACh levels have been found in microdialysis samples after induced status epilepticus in rats[46,47]. In addition, ACh concentrations at the synaptic cleft appear to reach the millimolar range during even normal synaptic transmission[48,49].

To determine whether the occurrence of seizures coincided with increased ACh levels, we utilized a recently developed acetylcholine sensor, iAChSnFr[50], to image ACh in the hippocampus during seizures. This allowed us to more precisely define the temporal resolution of ACh changes during seizures better than microdialysis. We first examined ACh levels in our electrically evoked seizure paradigm. We injected AAV vectors encoding iAChSnFr along with the red-shifted calcium sensor jRGECO into the hippocampus to enable simultaneous calcium and ACh imaging. Mice were implanted with an imaging window over the hippocampus along with a stimulating electrode in the perforant path and a recording electrode in dorsal CA1. A headbar was attached to enable imaging as they ran along a treadmill (Fig. 4a). During electrically evoked seizures, we found a significant increase in the ACh signal, mirroring the increase in calcium signal observed (Fig. 4b, c).

To examine whether similar ACh increases also occurred during spontaneous seizures, we performed simultaneous ACh and calcium imaging in IHKA mice. We injected AAV vectors encoding iAChSnFr along with jRGECO into the hippocampus at least 3 weeks after unilateral kainic acid injection, and a recording electrode was implanted in the hemisphere ipsilateral to kainic acid injection to detect seizures. Due to space constraints and the presence of hippocampal sclerosis on the ipsilateral side, vectors were transduced and imaging was performed on the contralateral side (Fig. 4d). Previous work has shown that seizures in the IHKA mouse model can spread and be detected on the contralateral side[51]. Larger increases in calcium signal during detected seizures were significantly correlated with greater increases in ACh signal (Fig. 4e). When we analyzed seizures that were associated with a significant increase in calcium on the contralateral side, we found a simultaneous increase in ACh signal (Fig. 4f), with the ACh signal closely tracking the calcium signal. Together, these results show that seizures lead to increased cholinergic signaling in the brain, a phenomenon that can be exploited for on-demand activation of engineered receptors to control neuronal hyperexcitability.

## Discussion

Engineered ligand-gated chloride channels offer a powerful approach to directly control neuronal activity. In particular, the BARNI eLGIC has

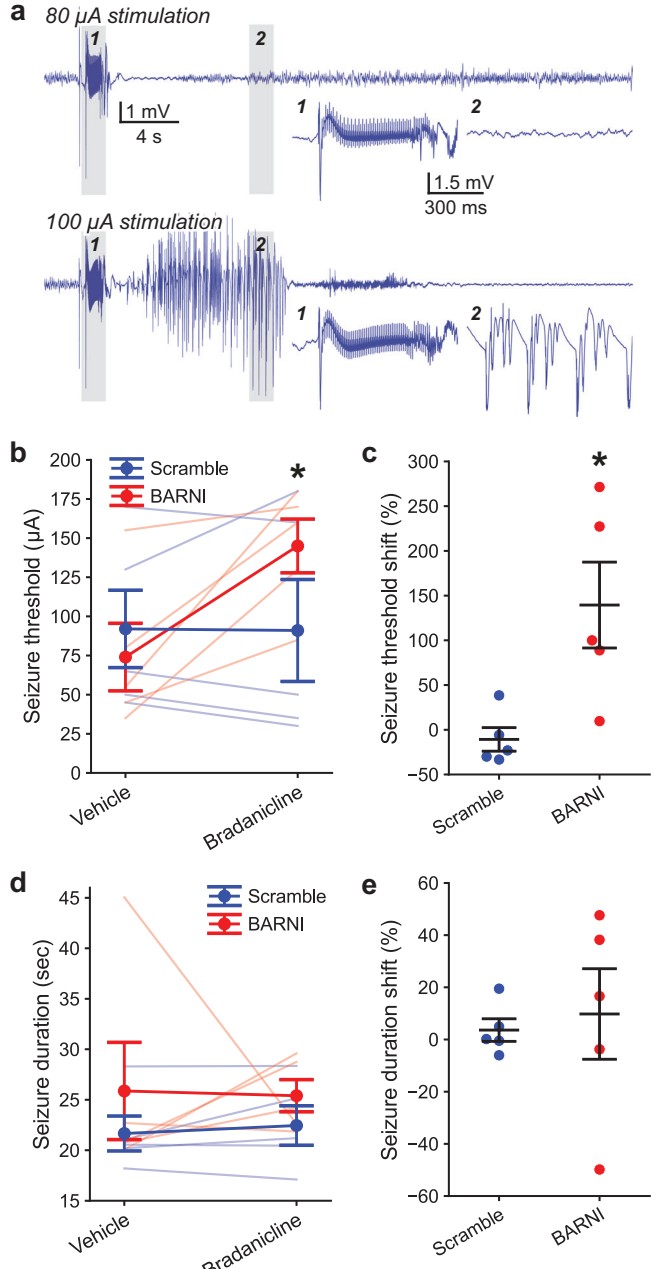

**Fig. 2 | BARNI channel activation increases evoked seizure thresholds.**
**a** Representative CA1 LFP responses to ipsilateral perforant pathway input electrical stimulation. Initial 1 s biphasic electrical stimulation *(inset 1)* of 80 μA failed to trigger a seizure, while subsequent 100 μA stimulation evoked sustained seizure activity with rhythmic bursting *(inset 2)*. **b** Bradanicline elevated seizure thresholds ($F_{(1, 8)} = 8.96$, $P = 0.017$), specifically related to an interaction for higher thresholds in BARNI animals ($F_{(1, 8)} = 9.48$, $P = 0.015$, two-way RM ANOVA). **c** Seizure threshold shifts, normalized to the vehicle response of each mouse, were likewise elevated in BARNI vs. Scramble animals ($t = 3.02$, $P = 0.017$, two-sided $t$-test). **d** Evoked seizure durations did not vary between animals with bilateral hippocampal expression of BARNI or Scramble constructs ($F_{(1, 8)} = 1.67$, $P = 0.233$), nor with bradanicline administration ($F_{(1, 8)} = 0.003$, $P = 0.958$) or due to an interaction of those two factors ($F_{(1, 8)} = 0.046$, $P = 0.835$, two-way RM ANOVA). **e** Seizure duration shifts, normalized to the vehicle response for each mouse, were similarly unchanged between vector groups ($t = 0.34$, $P = 0.739$, two-sided $t$-test). Bradanicline (100 mg/kg) was delivered i.p. 45 min prior to testing. In **b** and **d**, light lines show individual animal responses. Mean ± SEM, $n = 5$ mice per vector injection group. *$P < 0.05$. Multiple comparisons used two-sided Bonferroni correction. Source data are provided as a Source Data file.

several advantages that make it a promising candidate for development as a novel therapy. The α7 nAChR ligand-binding domain has been extensively studied, with known crystal structures of α7 nAChR homologs bound to various agonists[52,53]. α7 nAChRs themselves have been suggested as drug targets for various neurological disorders such as schizophrenia and Alzheimer's disease, and several clinically tested modulators of the receptor have been developed[54]. The BARNI-engineered channel, which is highly sensitive to the α7 nAChR-selective agonist bradanicline, is an exciting tool to enable effective control of seizures. Our results provide emerging evidence for the use of an inhibitory eLGIC, BARNI, as a potential therapeutic approach to treating epilepsy.

Our findings are consistent with previous work utilizing the DREADD-based system for chemogenetic silencing of neurons in mouse models of epilepsy. In these studies, i.p. administration of CNO led to a significant reduction in the number of observed seizures in various mouse models of epilepsy ranging from the IHKA model of TLE to a model of chronic focal neocortical epilepsy[26,55]. Notably, DREADD seizure suppression was found to last for several hours, even after a single administration of CNO, and despite the difference underlying the mechanism of the effect, we also observed prolonged seizure suppression with BARNI after i.p. injection of bradanicline. However, effector molecules used for DREADD-based approaches may act on a range of receptor types, leading to potential off-target effects[56]. Our approach takes advantage of the selectivity of bradanicline, which has 1000-fold greater affinity for the α7 subtype over the α4β2 subtype, the two most predominant nAChR subtypes in the brain, and has minimal interaction with other non-nicotinic receptor classes[41,57]. In addition, as previously mentioned, bradanicline has been evaluated through multiple phase 2 clinical trials and has been shown to be safe and well tolerated[37,38]. Indeed, we did not observe any substantial impacts of bradanicline administration, with or without BARNI expression, on memory-associated hippocampal theta, gamma, or ripple oscillations. However, our initial evaluation does not preclude more subtle effects on behavior, which should be further examined, including in the context of the potential coincident cognitive benefits of improved seizure control[58–61]. Interneurons, such as parvalbumin-expressing basket cells, are important drivers of memory-associated neuronal oscillations[62–64]. A potential avenue for further development of our chemogenetic platform is to target BARNI expression solely to excitatory neurons, which might provide even more effective seizure control while minimizing the potential for any cognitive impacts. While the PSAM eLGICs have been used in transgenic mice to study neurological disorders such as ALS and schizophrenia[65,66], our application of this approach to treat a disease phenotype in wild-type mice paves the way for its therapeutic potential in humans.

The role of cholinergic signaling itself in epilepsy has been unclear. Increases and decreases in both muscarinic and nicotinic AChR function have been reported from various animal epilepsy models and human patients[67]. Notably, we did not find any increase in seizures with bradanicline administration in Scramble-expressing animals, suggesting that activation of endogenous α7 nAChRs themselves does not have a detrimental effect on seizure susceptibility. Recent work has found that the firing activity of cholinergic neurons in the medial septum is reduced during kindled seizures[68]. However, seizures themselves could lead to the uncoupling of ACh release at presynaptic terminals from spiking activity at the soma. In addition, the sites of stimulation to elicit electrically evoked seizures in the previous study are different than ours (hippocampal CA3 versus perforant path). We found increased ACh signaling in both electrically evoked and

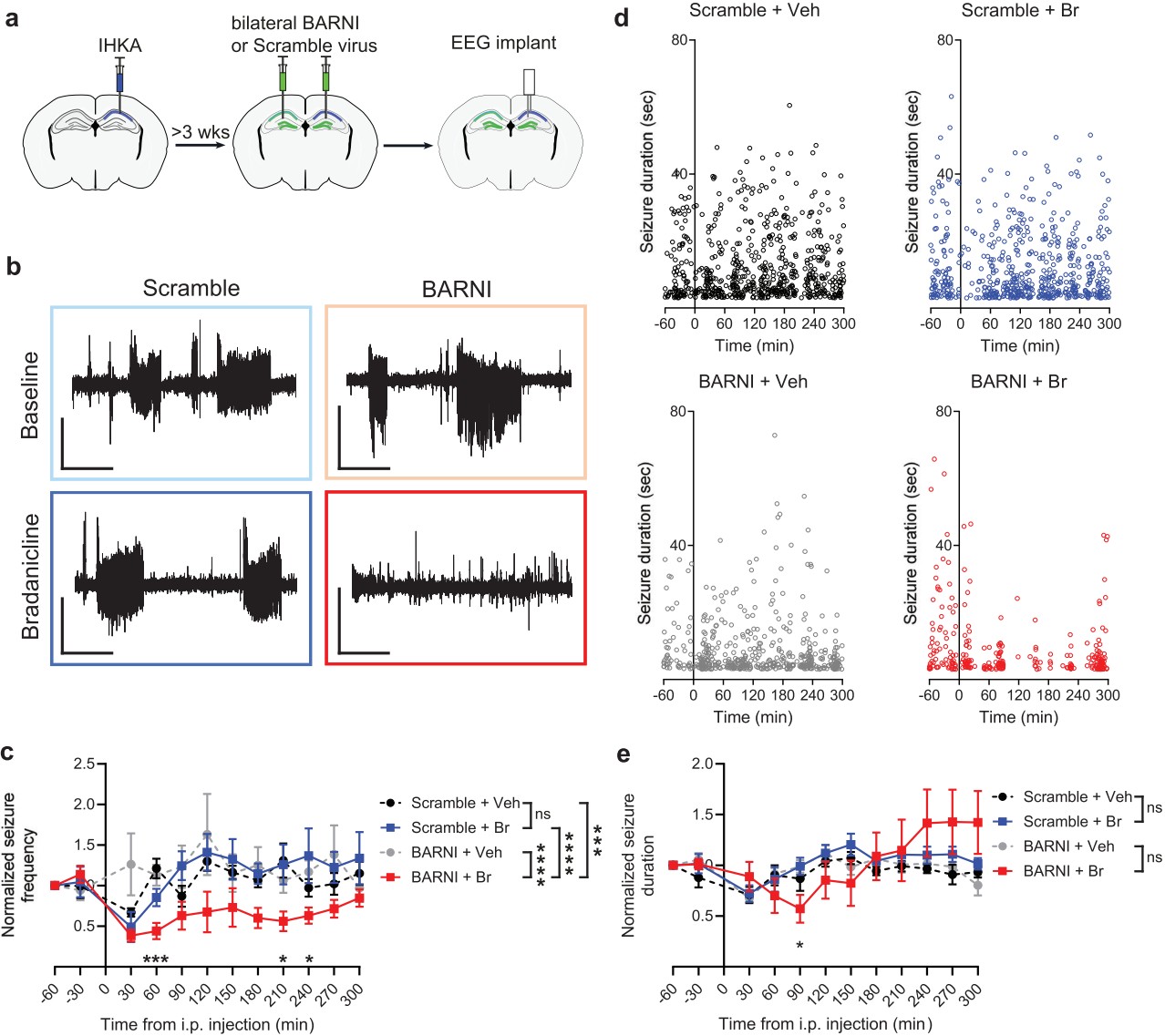

**Fig. 3 | BARNI channel activation decreases the frequency of spontaneous seizures. a** Experimental setup. Mice received unilateral dorsal intrahippocampal kainic acid (IHKA) injections. After at least 3 weeks for the development of chronic spontaneous seizures, mice received bilateral hippocampal injection of AAV virus encoding either BARNI or a Scramble control vector and implanted with EEG recording electrodes. **b** Representative EEG traces observed in mice expressing BARNI or Scramble control vector at baseline and after a single i.p. injection of bradanicline. Scale: 0.4 V, 40 s. **c** Frequency of spontaneous seizures decreased after a single i.p. injection of bradanicline (Br). 30-min bins, values normalized to seizure frequency observed one hour before treatment for each mouse ($F_{(33, 374)} = 2.137$, $P = 0.004$, two-way RM ANOVA, Time × Vector + Drug). **d** Representative time course of detected seizures in Scramble and BARNI

expressing mice after either vehicle (Veh) or bradanicline (Br) i.p. injection. **e** Brief change in the duration of spontaneous seizures observed after a single i.p. injection of bradanicline. 30-min bins, values normalized to seizure duration observed one hour before treatment for each mouse ($F_{(33, 374)} = 3.239$, $P < 0.0001$, two-way RM ANOVA, Time × Vector + Drug). Significance values on graphs are shown for comparison between vehicle and bradanicline treatment in BARNI-expressing mice at each time point using paired $t$-test. Significance values in legend are shown for comparison between vector expression and treatment groups using two-sided Tukey's multiple comparisons tests. Mean ± SEM, $n = 9$ Scramble, 10 BARNI mice. *$P < 0.05$, ***$P < 0.001$, ****$P < 0.0001$. Exact $P$-values are in Supplementary Table 1. Source data are provided as a Source Data file.

spontaneous seizures, suggesting that our findings are consistent across both acute and chronic models of epilepsy.

While chemogenetic systems have often prioritized the development of receptor/drug combinations that minimize interaction with endogenous molecules, maintenance of endogenous ligand binding to compounds that become elevated during seizures could also pave the way for more effective therapy in pathological conditions such as epilepsy. Interestingly, we saw a significant decrease in the cumulative number of seizures from the expression of the BARNI receptor alone. This effect is similar to what is seen with other engineered inhibitory channels activated by either glutamate or immediate early gene

expression[13,69]. Our results suggest that the increase in ACh levels seen during seizures could provide an additional additive effect on receptor activation in combination with exogenous ligand binding, or can mediate on-demand receptor activation when levels of exogenous ligand are low.

## Methods
### Subjects
Adult male C57BL/6 mice (Jackson Laboratory for Stanford University, Charles River for CODA Biotherapeutics) were used for all experiments, except for embryos from timed pregnant female mice used to

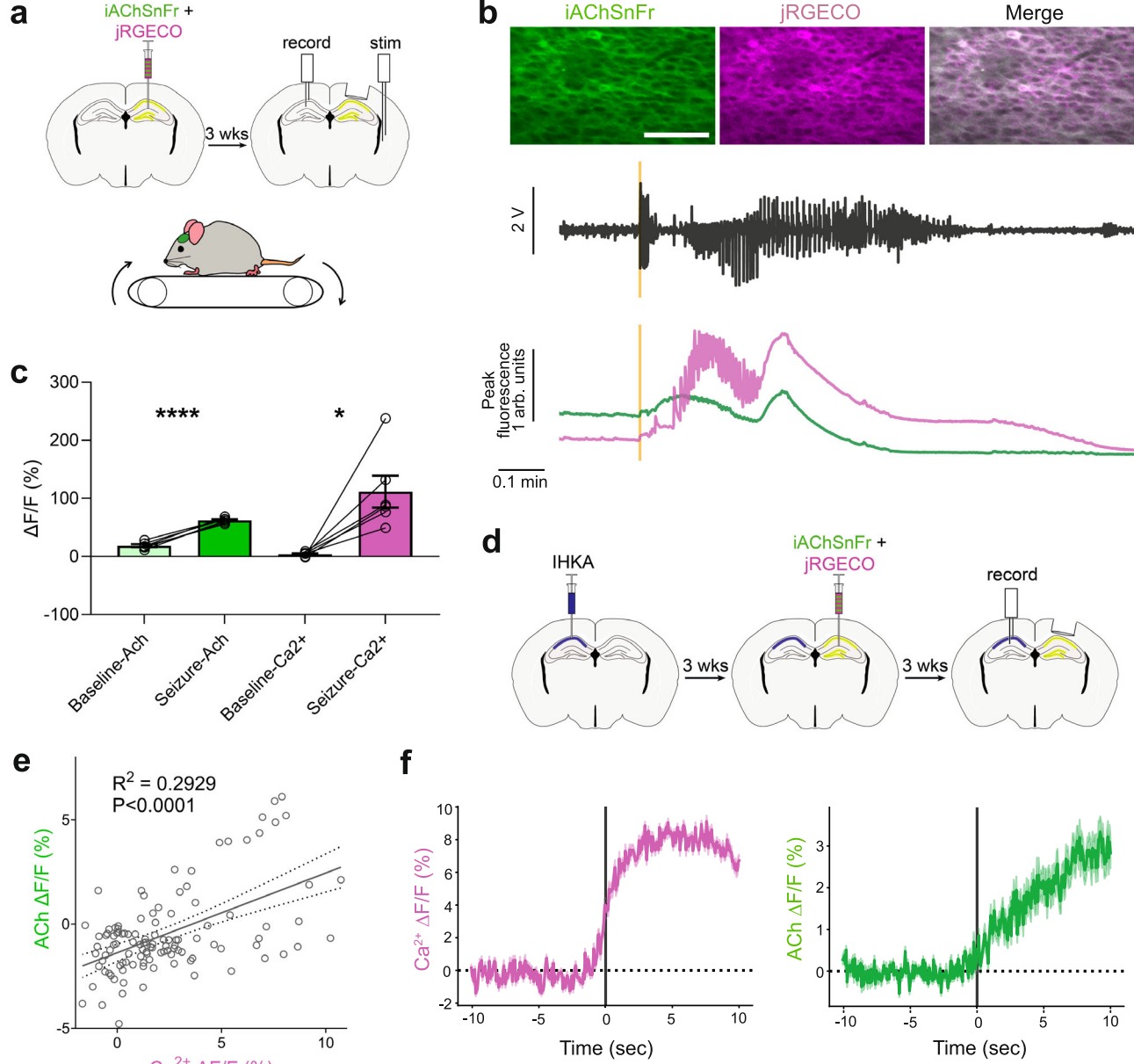

**Fig. 4 | Increased ACh levels during seizures. a** Experimental setup for imaging of electrically evoked seizures. Mice were injected with AAV viruses encoding the ACh sensor iAChSnFr and the red-shifted calcium sensor jRGECO, then implanted with a stimulation electrode in the perforant path and a recording electrode in dorsal hippocampus. Imaging was performed while mice were running on a treadmill. **b** *Top*, Representative images of iAChSnFr (green), jRGECO (magenta), and merged signals. Scale bar: 100 μm. *Bottom*, Representative local field potential (LFP), ACh and calcium sensor signal recorded during evoked seizures. Yellow line denotes start of the perforant path stimulation. **c** Increased ACh and calcium signal fluorescence (F) was observed during electrically evoked seizures. Paired two-sided *t*-test. Mean ± SEM, *n* = 6 seizures from 4 mice. *P = 0.0124, ****P < 0.0001. **d** Experimental setup for imaging of spontaneous seizures in intrahippocampal kainic acid (IHKA) mice. Mice received unilateral injection of kainic acid into the

dorsal hippocampus. After at least 3 weeks for the development of chronic spontaneous seizures, mice received hippocampal injection of AAV viruses encoding the ACh sensor iAChSnFr and the red-shifted calcium sensor jRGECO in the hemisphere contralateral to kainic acid injection and implanted with a recording electrode in the dorsal hippocampus in the hemisphere ipsilateral to kainic acid injection. **e** Significant correlation between ACh and calcium signal changes during spontaneous seizures. Responses are average signal changes in the 10 s after the start of the seizure. Linear regression line is shown with 95% confidence bands. *n* = 5 mice, 15 imaging sessions, 112 seizure events. **f** Average calcium and ACh responses aligned to the start of seizure for events that resulted in significant rises in calcium in the contralateral hemisphere across multiple mice and imaging sessions. Mean ± SEM, *n* = 3 mice, 4 imaging sessions, 21 seizure events. Source data are provided as a Source Data file.

prepare cultured hippocampal neurons. All procedures were carried out in accordance with the National Institutes of Health guidelines for animal care and use and were approved by the Administrative Panel on Laboratory Animal Care of Stanford University (Protocol #30183) or the Institutional Animal Care and Use Committee at Coda Biotherapeutics. Mice were group housed (2–5 per cage) except during EEG recordings, received ad libitum access to food and water, and were

maintained on a 12 hr light/dark cycle throughout the study under standard housing conditions (21 ± 2 °C; 50 ± 15% humidity).

### Vector production
The BARNI receptor used in the current study is the CODA71 variant developed by Coda Biotherapeutics. AAV6 and AAV9 vectors carrying either the BARNI receptor expression cassette or

the scrambled control cassette were produced in HEK293 cells (AAV-293, Agilent) using standard methods. The vectors were titered using ddPCR and purity was confirmed using SDS–PAGE.

## Cultured cell electrophysiology

Mouse hippocampal neurons (mHC) were harvested at P0, dissociated with trypsin, and cultured on poly-L-lysine coated glass coverslips in media containing: 73.5% neurobasal, 23.2% minimal essential media, 2% B-27, 0.04% glutamax (all Thermo Fisher) and 1.25% FBS (Hyclone). Samples were pooled from multiple embryos, generating cultures with a mix of neurons from male and female sources. At day in vitro (DIV) 3, mHC cells were transfected with $3 \times 10^8$ vg of AAV6 vector containing either an hSyn-BARNI-HA-IRES-GFP expression cassette or an hSyn-Scramble-HA-IRES-GFP scrambled control cassette. Starting at DIV7, 5-fluoro-deoxyuridine was added to mHC cultures to inhibit glial growth. All mHC experiments were conducted between DIV13 and 16.

Electrophysiology recordings used cells plated on coverslips that were visualized on an inverted fluorescence microscope (Olympus), while in room temperature extracellular solution (ECS) containing (in mM): 140 NaCl, 4 KCl, 1 $MgCl_2$, 2 $CaCl_2$, 10 HEPES, and 10 glucose (pH-adjusted to 7.3 with NaOH, osmolarity 305–315 mOsm), was perfused at 0.5 ml/min. Recording pipettes were pulled from thick-walled borosilicate capillary glass (Sutter, BF150-86-10) using a P97 puller (Sutter Instruments) and had a 3–6 MΩ tip resistance. For mHC recordings pipettes contained (in mM): 115 K-gluconate, 13 KCl, 1 $MgCl_2$, 5 ATP-Mg, 10 HEPES, and 0.5 EGTA (pH adjusted to 7.2, osmolarity 290–295 mOsm).

For dose-response experiments, whole-cell recordings were performed on mHC cells voltage-clamped at −60 mV. Doses of bradanicline (Targacept, TC-5619; CAS: 639489-84-2) or acetylcholine chloride (Sigma, A6625) were applied using an 8-line reservoir (AutoMate Scientific) in combination with an 8-channel zero-dead-volume perfusion pencil (AutoMate Scientific). Drug was applied for 1–10 s until a peak shift in holding current was observed, with >2 min washes between all drug applications. Drug doses were applied in increasing half-log increments until the peak current amplitude plateaued. Baseline holding currents were subtracted from the response peak to find the drug-induced current amplitude for each concentration. Drug-induced current amplitudes were normalized to the peak response within each cell. Normalized current amplitudes were then averaged across cells and the data were fit to dose–response curves using nonlinear regression models ($Y = 100/(1 + EC_{50}/X)^{Hillslope}$). Capacitance and series resistance were compensated for all recordings, which were collected at 3 kHz with a 1 kHz Bessel filter.

Intrinsic mHC neuron properties were monitored in whole-cell current clamp recordings where no holding current was applied. Periodic series of 500 ms current steps (−200 to 700 pA) were used to longitudinally track neuronal properties. Input resistance ($R_{in}$) was calculated from the change in steady-state membrane potential, averaged across the four hyperpolarizing current injection steps. Rheobase values were the minimal current during current step delivery that evoked an action potential (AP). If rheobase currents were ambiguous when APs appeared to be graded in amplitude, a polar plot of the trace clarified the rheobase. Measurements were collected for an initial 1 min baseline period, before a 20 min application of bradanicline (300 nM) followed by 3 min of drug washout. NBQX (10 μM) and AP5 (50 μM) were included in the bath throughout the recordings to block excitatory synaptic transmission. Hippocampal spontaneous APs were evaluated in separate current clamp recordings, with no holding current applied. As with dose–response experiments, half-log increment doses of bradanicline were applied for 90 s at each dose, and the APs occurring within a stable 60 s time window during that period were counted. AP frequency was calculated as the number of APs occurring per second of each drug application period and then normalized to the firing frequency of each cell during the baseline

period. Data were acquired at 10 kHz, with Bessel filter at 2 kHz and series resistance compensation at 100%. Data were excluded from recordings where cells were lost in the middle of recordings, perfusion issues were observed, or from BARNI-expressing cells showing <40% change in input resistance to maximal drug application.

Recordings were acquired with an Axopatch 200B amplifier and Digidata 1550B (Molecular Devices). Data was collected using pClamp software (Molecular Devices). Traces were either analyzed in Clampfit (Molecular Devices) or using custom Python code, with statistics calculated in GraphPad Prism.

## Slice electrophysiology

Mice received an initial stereotaxic injection bilaterally into CA1 (M/L: ±1.5; A/P: −2.3; D/V: −1.35 mm) when 3–4 weeks old. Each mouse received 400 nL injections of an AAV9 vector containing either an hSyn-BARNI-HA-IRES-GFP expression cassette ($2.6 \times 10^{12}$ vg/mL) or an hSyn-Scramble-HA-IRES-GFP scrambled control cassette. At 2–5 weeks following initial injection, mice were deeply anesthetized by Ketamine/Xylazine and then transcardially perfused with an ice-cold protective recovery solution containing (in mM): 92 NMDG, 26 $NaHCO_3$, 25 glucose, 20 HEPES, 10 $MgSO_4$, 5 Na-ascorbate, 3 Na-pyruvate, 2.5 KCl, 2 thiourea, 1.25 $NaH_2PO_4$, 0.5 $CaCl_2$, titrated to a pH of 7.3–7.4 with HCl[70]. Coronal slices (300 μm) containing the hippocampus were cut in ice-cold protective recovery solution using a vibratome (VT1200S, Leica Biosystems). Brain slices were then incubated in 35 °C protective recovery solution for 12 min. Subsequently, brain slices were maintained in room temperature aCSF consisting of (in mM): 126 NaCl, 26 $NaHCO_3$, 10 glucose, 2.5 KCl, 2 $MgCl_2$, 2 $CaCl_2$, 1.25 $NaH_2PO_4$. All solutions were equilibrated with 95% $O_2$/5% $CO_2$.

Intracellular recordings were performed in a submerged chamber perfused with oxygenated aCSF at 3 ml/min and maintained at 33 °C by a chamber heater (BadController V, Luigs and Neumann). CA1 neurons were visualized using DIC illumination on an Olympus BX61WI microscope (Olympus Microscopy) with an sCMOS camera (Flash 4.0 LT+, Hamamatsu). Epifluorescence illumination from an LED lamp (Solis-3C, Thorlabs) was used to identify GFP-positive transfected neurons. Recording pipettes were pulled from thin-walled borosilicate capillary glass (King Precision Glass) using a P97 puller (Sutter Instruments) and were filled with (in mM): 126 K-gluconate, 10 HEPES, 4 KCl, 4 ATP-Mg, 0.3 GTP-Na, 10 phosphocreatine (pH-adjusted to 7.3 with KOH, osmolarity 290 mOsm), as well as 0.2% biocytin. Pipettes had a 3–5 MΩ tip resistance.

Whole-cell recordings were performed on GFP-positive CA1 neurons in the dorsal hippocampus (A/P: −1.5 to 2.4 mm). Pipette capacitance was neutralized for all recordings and holding current was adjusted so that all cells began recordings with an initial membrane potential of −65 mV. Intrinsic firing properties were assessed during current injection steps (−100 to 350 pA, 1 s), before initial drug application. Action potential threshold was the voltage where the d$V$/d$t$ prior to a detected event first exceeded 3 times the standard deviation. Width was the time an action potential, resampled at 100 kHz, exceeded the half-height between threshold and peak voltages, and cells with a width 1 ms or less were considered to be putative fast-spiking interneurons[71,72]. Action potential properties were only measured in the first spike evoked by a depolarizing current for each neuron.

Neuronal properties were then assessed longitudinally, across 5 s sweeps, featuring repeated current injection patterns including a brief hyperpolarizing current step (−100 pA, 200 ms), followed shortly later (300 ms) by a linearly ramping current delivery (−150 to +500 pA, across 2 s). Input resistance ($R_{in}$) was calculated from the change in steady-state membrane potential resulting from hyperpolarizing current injections. Resting membrane potential (RMP) was measured as the average value during the period (500 ms) in each sweep prior to any current injection.

All APs were counted during the ramping current delivery and the rheobase value was the current being delivered when the first AP of

each sweep was evoked. In sweeps with no detected APs, a rheobase current of 500 pA was assigned. Data was excluded from cells where no APs were detected during the initial sweep or when $R_{in}$ was >500 mΩ, as well as individual sweeps where the RMP was >−40 mV. Responses were tracked during bath application of bradanicline (150 nm, Targacept) and α-Bungarotoxin (0.25 μM, Abcam). Data was collected from $n = 19$ cells/8 animals BARNI, $n = 16$ cell/9 animals scramble. Data were acquired in pClamp software (Molecular Devices) using a Multiclamp 700B amplifier (Molecular Devices), low-pass filtered at 2 kHz, and digitized at 10 kHz (Digidata 1440A, Molecular Devices). Data analysis was performed using custom-written Python scripts.

A subset of slice electrophysiology recordings from CA1 neurons was independently replicated at CODA using similar approaches. Stereotaxic injections were made at comparable coordinates in CA1 (M/L: +1.6; A/P: 2.0; D/V: −1.4 mm) with 400 nL of $1.6 \times 10^{12}$ vg/mL of AAV9 vector. Mice were also perfused with NMDG-based protective recovery solution, prior to embedding in low-melting point agarose and cutting into coronal slices (275 μm) on a Compresstome VF-310-0Z (Precisionary Instruments). Slices were next incubated in NMDG solution at 34 °C for 10 min, then maintained for the rest of the experiment at RT in an alternative ACSF formulation, containing (in mM): 127 NaCl, 25 NaHCO3, 25 glucose, 2.5 KCl, 2 CaCl2, 1 MgCl2, 1.25 NaH2PO4, bubbled continuously with 95% O2 and 5% CO2. Similar current clamp responses to ramping current injections were performed using a slightly different pipette internal solution containing (in mM): 115 K-gluconate, 13 KCl, 1 MgCl2, 5 ATP-Mg, 10 HEPES, and 0.5 EGTA (pH adjusted to 7.2, osmolarity 290–295 mOsm). Data were acquired in pClamp software (Molecular Devices) using a Multiclamp 700B amplifier (Molecular Devices) and digitized on a Digidata 1550B Plus Humsilencer (Molecular Devices). Data analysis was performed using custom-written Python scripts.

## Electrically evoked seizure thresholds

During an initial surgery, mice were stereotaxically injected bilaterally into CA1 and DG (M/L: ±1.5; A/P: −2.3; D/V: −2.0, −1.6 and −1.35 mm) when 5–7 weeks old. Each injection site received 200 nL of an AAV9 vector containing an hSyn-BARNI-HA-IRES-GFP expression cassette ($2.6 \times 10^{12}$ vg/mL) or in control animals, an hSyn-Scramble-HA-IRES-GFP scrambled cassette. Bipolar, stainless steel, recording electrodes (MS303-A, P1Technologies), twisted to provide a 0.5–1 mm tip separation, were placed in the right hippocampus (M/L: 1.5; A/P: −2.3; D/V: −1.5 mm), as well as the ipsilateral perforant pathway (M/L: 2.5; A/P: −4.1; D/V: −2.9 mm), and a head bar was secured to the skull.

Mice were habituated to head-fixed running on a linear treadmill for two 15 min sessions prior to testing. During seizure threshold testing sessions at 3–5 weeks after initial virus injection, mice were head-fixed on a linear treadmill while hippocampal activity was recorded with a differential AC amplifier (Model 1700, A-M Systems), filtered between 1 Hz and 20 kHz and digitized at 5 kHz using a NIDAQ data acquisition card (National Instruments), paired with custom written MATLAB data acquisition scripts. Biphasic, constant current, electrical stimulation was delivered to the perforant pathway in 1 s trains of 1 ms pulses at 60 Hz from an isolated pulse stimulator (Model 2100, A-M Systems). Electrical stimulation was delivered once every 100 s at gradually increasing amplitudes ranging from 10 to 400 μA in 10–30 μA steps (10–100:10 μA steps; 100–240:20 μA steps; >240:30 μA steps) until an electrographic hippocampal seizure was observed. Evoked seizures were considered sustained periods (≥6 s) of elevated amplitude hippocampal signal, characterized by rhythmic bursting at 2-4 Hz along with a higher frequency component (20–50 Hz), which began acutely (<10 s) after electrical stimulation ceased.

On seizure threshold testing days, mice received an i.p. injection of bradanicline (100 mg/kg) or saline 45 min prior to the start of recording. Seizure thresholds were measured twice for each injection condition, in a shuffled order, and always separated by >24 h between trials. Seizure thresholds and durations were tracked for each experiment and then averaged, within animal, across replicate trials. Data was collected from $n = 5$ animals per BARNI and scramble group.

## Local field potential recordings

Mice received similar viral injections matching the electrically-evoked seizure threshold experiments. Linear 4-channel silicon probes (Q1 × 4-3mm-100-177, NeuroNexus), with 100 μm spacing between recording sites were implanted in the right hippocampus (tip at M/L: 1.5; A/P: −2.3; D/V: −1.5 mm), along with a brain surface ground wire (M/L: 2.5; A/P: −4.1 mm). Prior to the first recording session, mice were twice habituated to the open field arena (30 × 24 cm) for 10 min. Recording sessions lasting 1 h, where the mice were allowed to freely explore, occurred at least 2 weeks after virus injection, with data acquired at 30 kHz using an Open Ephys acquisition system and RHD2132 unipolar-input recording headstages (Intan Technologies). Mice received a saline injection at the start of each recording and an i.p. injection of bradanicline (100 mg/kg) 30 min later.

Active periods were assessed based on continuous elevated head-mounted 3-axis accelerometer signals of >5 s. LFP power within the theta (4–10 Hz), low gamma (20–55 Hz) and high gamma (55–90 Hz) bands was measured using a multitaper spectral analysis only during active periods, selecting the data from the recording channel with the greatest theta power from each animal. Sharp wave-ripples (ripples) were defined as high-frequency oscillatory periods, evaluated only while animals were inactive. Ripples were detected in periods where the filtered (90–200 Hz) LFP signal amplitude exceeded 5 standard deviations beyond the overall background signal amplitude variability of the trace. Ripples were considered to extend to contiguous periods of LFP > 1 standard deviation above noise. Intra-ripple frequency was the frequency of the oscillations composing ripples and was calculated independently for each ripple event using Welch's method, while the average amplitude of each ripple was measured from a Hilbert transform. Ripples were analyzed in the recording channel with the greatest ripple amplitude from each animal.

## Measuring bradanicline plasma and brain levels

Mice received 100 mg/kg of bradanicline via either oral gavage (10 mg/ml) or i.p. injection (20 mg/ml). Plasma and brains were collected from animals at 0.5, 1, 2 and 4 h timepoints after dosing ($n = 3$ mice/group). Mice were euthanized and 0.3 ml of blood was collected via a cardiac puncture into K2EDTA-coated vacutainer tubes, which were centrifuged at $4000 \times g$ for 5 min at 4 °C to prepare plasma. Mice then underwent cardiac perfusion with ice-cold saline prior to brains being collected and snap-frozen. All samples were stored at −75 °C prior to analysis.

A range of standards for generating calibration curves for plasma analysis (0.5, 1, 2, 5, 10, 50, 100, 500, 1000 ng/ml) was produced by diluting bradanicline in a mixture of plasma from control mice, acetonitrile, and water (4:1:1 ratio, respectively). Quality control samples (1, 2, 50, and 800 ng/ml) were independently prepared, on the day of analysis, using the same approach. Experimental plasma samples were likewise diluted 4:1:1 with acetonitrile and water. Brain samples were homogenized in a mixture of ultrapure water with 10% acetonitrile and 0.1% formic acid, adding 3 ml of solution per gram of brain tissue. As with plasma samples, homogenized brain tissue was diluted 4:1:1 with acetonitrile and water. Likewise, calibration curve (1, 2, 4, 20, 100, 200, 1000, 2000, 4000 ng/ml) and quality control (1, 2, 50, and 1600 ng/ml) samples were prepared as for plasma but using homogenized brain tissue from control mice.

All experimental, calibration, and quality control samples were prepared for analysis by combining 15 μl of the prepared solution with 200 μl acetonitrile to precipitate proteins and an internal standard of propranolol (500 nM). Samples were vortexed for 30 s, prior to

$1541 \times g$ centrifugation at 4 °C for 15 min. Supernatant was diluted 3-fold with water and 9 μl of the final sample was injected into the LC–MS/MS system for quantitative analysis. Bradanicline detection and quantitation were performed using an API5500 LC-MS/MS (Applied Biosystems) with either a Phenomenex Synergi 2.5 μm Polar-RP 100 Å (50 × 3 mm) or a Raptor Biphenyl 2.7 μm (50 × 2.1 mm) column. For the mobile phase, Solution A was 5% acetonitrile in water (with 0.1% formic acid) and Solution B was 95% acetonitrile in water (with 0.1% formic acid). The flow rate was 0.6 ml/min.

## EEG recordings

All surgeries were conducted under aseptic conditions using a small-animal stereotaxic instrument (AngleTwo, Leica Biosystems Inc.), UMP3 ultramicropump (WPI), and Hamilton syringes (65458-01, 65457-01). Mice were anaesthetized with isoflurane (5% for induction, 1.5–2.0% after) in the stereotaxic frame for the entire surgery and their body temperature was maintained using a heating pad. For epilepsy induction, 7–9-week-old mice were stereotaxically injected with kainic acid (40 nl, 20 mM in saline; Tocris) into the right dorsal hippocampus (M/L: −1.25; A/P: −2.0; D/V: −1.6 mm). Mice were then put back into their home cages for at least 3 weeks for the development of chronic spontaneous seizures. Afterwards mice were bilaterally injected with 200 nL of AAV9-hSyn-BARNI-HA-IRES-GFP ($2.6 \times 10^{12}$ vg/ml) or AAV9-hSyn-Scramble-HA-IRES-GFP ($2.2 \times 10^{12}$ vg/ml) at each of three depths at a rate of 100 nl/min into the hippocampus (M/L: ±1.5; A/P: −2.3; D/V: −1.35, −1.6 and −2.0 mm). On the same day right after vector injection mice were implanted with a bipolar EEG depth electrode twisted with 0.5–1 mm tip separation (P1Technologies, MS303-2) into the right kainic acid injected hemisphere (M/L: −1.5; A/P: −2.3; D/V: −1.55 mm). Afterward, mice were connected to a 24/7 video-EEG monitoring system to record the occurrence of seizures: briefly, EEG recording electrodes for kainic acid injected animals were connected to an electrical commutator (P1Technologies, SL2C/SB) routed to an amplifier (BrownLee 410; Automate Scientific, Inc.), and in turn connected to a digitizer (National Instruments USB-6221; National Instruments) and a computer running custom MATLAB recorder and seizure detection software.

After at least 3 weeks to allow for vector expression, mice were then i.p. injected with either bradanicline (100 mg/kg) or saline vehicle. Mice received at least two days of injection with either bradanicline or saline. The frequency and duration of seizures were then analyzed before and after drug or vehicle injection. For semi-automatic analysis of spike clusters (i.e., seizures), the custom MATLAB program used different detection criteria provided by the experimenter for local field potential spikes (including filtering, amplitude threshold, width and template matching), local field potential spike clusters (including inter-spike-interval, inter-cluster-interval and minimal duration) and artifact rejection (including different filters and signal features), which were combined using Boolean logic. The experimenter verified and if necessary corrected all processed files on their detection accuracy of seizure starts and ends. Spike clusters >3 s in duration with an inter-spike interval of <1 s were included as seizures. Similar results were obtained with analysis using a longer 6-s seizure threshold.

## In vivo ACh imaging

Similar to the electrically-evoked seizure experiments, mice were stereotaxically injected bilaterally into CA1 and DG (M/L: ±1.5; A/P: −2.3; D/V: −2.0, −1.6 and −1.35 mm) when 5–7 weeks old. Each injection site received 200 nL of a 1:4 mixture of AAV1-hSyn-iAChSnFR (transfer plasmid #137950, Addgene; gift from Loren Looger) and AAV1-Syn-NES-jRGECO1a-WPRE-SV40 (transfer plasmid #100854, Addgene; gift from Douglas Kim & GENIE Project). After allowing 2 weeks for vector-mediated expression of the transgene and recovery from the injection, the cortex above the injection site was aspirated and a stainless steel cannula with attached coverglass was implanted over the hippocampus, followed by a stainless steel headbar as described earlier[73–75].

One week later, bipolar, stainless steel, recording electrodes (MS303-A, P1Technologies), twisted to provide a 0.5–1 mm tip separation, were placed in the right hippocampus (M/L: 1.5; A/P: −2.3; D/V: −1.5 mm), as well as the ipsilateral perforant pathway (M/L: 2.5; A/P: −4.1; D/V: −2.9 mm).

Head-fixed mice were imaged using a resonant scanner 2-photon microscope (Neurolabware), equipped with a pulsed IR laser (Mai Tai, Spectra-Physics), gated GaAsP PMT detectors (H11706P-40, Hamamatsu), and 16x objective (0.8 NA WI, Nikon). 2-photon image acquisition was controlled by a Scanbox (Neurolabware) system, which also synchronized behavioral video acquisition (Mako, Allied Vision), treadmill speed monitoring, and field potential recording via a DAC (National Instruments) to the imaging frames. Imaging was performed mainly of the pyramidal layer of area CA1 of the hippocampus, with some images containing cells from the oriens layer. Baseline signal 2-photon and hippocampal LFP were recorded while head-fixed mice were allowed to spontaneously run or rest on an uncued linear treadmill. Then, as in the electrically evoked seizure experiments, escalating intensity electrical currents were periodically delivered to the perforant pathway until an electrographic seizure was detected in CA1. In IHKA animals, imaging and LFP recording were performed in 20–30 min sessions. Data was collected at 15 fps with the laser tuned to 1000 nm to simultaneously sample both sensors with low crosstalk between green and red channels.

Calcium imaging data were processed and analyzed using Python scripts. Movies were initially motion corrected by rigid translation, followed by non-rigid correction (HiddenMarkov2D function of sima)[76]. Binary regions of interest (ROIs) were selected in a semi-automated manner to include single-cell bodies. For the initial automated detection, movies were divided into segments of 100 frames each, the average intensity projection of each segment was computed and the resulting resampled movie was used for detection. For the detection of pyramidal cells, the PlaneCA1PC method of sima was run on the inverted resampled movie, which resulted in the detection of the hollow nuclei of cells. These ROIs were filtered based on size, and binary dilation was performed to include the cytoplasm around the nuclei. In a subsequent step, ROIs were detected in the non-inverted resampled movie, filtered based on size and those that did not overlap with existing ROIs were added to the set. This approach detected a large number of ROIs that corresponded well to individual cells, although some cells may be represented by more than one ROI and some ROIs may contain overlapping parts of multiple cells. ROIs for putative interneurons, which had larger cell bodies than pyramidal cells, were manually drawn. Next, the fluorescence intensity traces were extracted for each ROI, processed to obtain DF/F traces[75]. A time-dependent baseline was computed by fitting a third-degree polynomial on the trace after applying temporal smoothing and ignoring peaks, periods of running, and the beginning and end of the recording.

For analysis of ACh and calcium responses during evoked seizures, baseline measurements were taken 30 s before the start of electrical stimulation and compared to the peak response during the seizure. For spontaneous seizure experiments, seizures were first identified using the criteria for seizure detection described above (>3 s duration and 1 Hz frequency). Next, a response metric was used to select seizures that coincided with significant increases in calcium DF/F in the contralateral hemisphere. This response metric was based on the peak signal during the 10 s after a time point, normalized to a baseline taken 10 s before the time point. Significant events in each session were selected if their seizure response was >95% of responses calculated at randomly selected time points. Mean seizure responses were aligned based on the start of the event at time 0. Seizures without a significant calcium response were excluded from the analysis, as these can correspond to events not spreading to the contralateral CA1.

## Perfusion, histology, and imaging

Following EEG experiments, animals were euthanized at least one week after the last i.p. injection, by being deeply anesthetized with a mixture of ketamine and xylazine (80–100 mg/kg ketamine, 5–10 mg/kg xylazine; intraperitoneal) and transcardially perfused with 10 mL of 0.9% sodium chloride solution followed by 40 ml of cold 4% PFA dissolved in phosphate buffer solution. The excised brains were held in a 4% PFA solution for at least 24 h before being sectioned into 60 µm slices using a vibratome (Leica VT1200S; Leica Biosystems Inc.). Sections were screened and chosen based on the area of greatest GFP expression, then mounted on glass slides and cover-slipped using Vectashield Antifade Mounting medium (Vector Laboratories). Imaging was performed on a Zeiss LSM 800 confocal microscope using a 20x objective. For each animal, 3–5 regions of interest were imaged for each region corresponding to CA1 and dentate gyrus in both hemispheres ipsilateral and contralateral to kainic acid injection. For each region of interest, a z-stack of 5–7 images was taken. All other imaging conditions were conserved across all regions and sections imaged.

## Statistical analysis

Graphs and statistical analyses were generated using GraphPad Prism or Python (with Pandas, Seaborn, Scipy, Statsmodels, and Pingouin packages). Seizures were binned into 30 min intervals before and after drug injection. For each mouse, seizure frequency and duration were normalized to values in the first 30 min bin starting from one hour prior to drug or vehicle injection. Analysis was done using a paired $t$-test comparing drug vs vehicle conditions at each time point. For analysis of the cumulative number of seizures during the baseline recording period, a mixed model utilizing a compound symmetry covariance matrix and fit using Restricted Maximum Likelihood was used with Geisser–Greenhouse correction. One Scramble-expressing mouse did not have a stable EEG signal until well into the baseline recording period and was excluded from the baseline analysis. Histology image quantification was performed using ImageJ. Intensity values were based on maximum intensity projection values of the whole z-stack. For each image, the area of analysis was chosen based on the freehand selection of cell bodies. Statistical analysis was performed on intensity values normalized to the area of the region analyzed. Sex was not considered in the current study design and a sex-based analysis was not performed, with our current data being restricted mostly to male samples.

## Reporting summary

Further information on research design is available in the Nature Portfolio Reporting Summary linked to this article.

# Data availability

Plasmid sequences have been deposited into GenBank, under accession number OR944653. The data used to generate all plots are accessible in the provided Source Data file. Underlying datasets have been deposited in the Zenodo database: https://doi.org/10.5281/zenodo.10223431[77]. Source data are provided with this paper.

# Code availability

Analysis code has been deposited in the Zenodo database: https://doi.org/10.5281/zenodo.10223431[77].

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

## Acknowledgements

CODA Biotherapeutics, Inc. provided reagents and funding for these experiments. We thank Gordon Wang and the Stanford Neuroscience Microscopy Service, supported by NIH grant NS069375, for providing microscopy support. We thank Edward Yeh, Nadia Kahn, Angela Liu, Kathy Anh Lam, Srijana Balasundar, Susan Catalano, and Sandra Linder for relevant discussions and technical support.

## Author contributions

Q.-A.N., P.M.K., J.H. and C.L.P. performed ex vivo and in vivo experiments. C.X., K.N.B., J.I. and J.T.B. performed in vitro and ex vivo experiments. B.D., J.S.F. and T.G. contributed to the in vivo imaging analysis, seizure threshold monitoring experiments, and semi-automated detection of seizures. A.K. and G.S.D. designed the development of the BARNI receptor and experiments for initial characterization. Q.A.N., P.M.K. and I.S. designed ex vivo and in vivo experiments. Q.-A.N. and P.M.K. wrote the manuscript with input from all authors.

## Competing interests

Q-.A.N., P.M.K., J.H., B.D., J.S.F., T.G., C.L.P. and I.S. declare no competing interests beyond utilizing reagents and funding from CODA Biotherapeutics for the experiments described in the paper. C.X., K.B., J.I., J.T.B., A.K. and G.S.D. were employees of CODA Biotherapeutics. Patent #10538571 has been issued to CODA Biotherapeutics on compositions and methods for modulating the activity of cells using engineered receptors, polynucleotide-encoded engineered receptors, and gene therapy vectors comprising polynucleotides encoding engineered receptors.
