## [Peer Review File · Nature Communications]

REVIEWER COMMENTS

Reviewer #1 (Remarks to the Author):

With much interest I have read this paper.

The authors developed an alternative PSAM receptor (but strangely the names PSAM-PSEM are nowhere mentioned in the paper) and showed suppression of seizures in the intrahippocampal kainic acid mouse model.

I have a couple of comments:

- Now the authors performed a single administration of the drug but what about repeated treatment?

- very few microscopy images are shown, especially regarding the expression of the acetylcholine biosensor and the calcium indicator. I think an illustrative image should be shown. The authors did perform 2-photon imaging. I am a bit worried about the fact that the serotypes for expression of the Ach and calcium sensor was the same. Is there no competition for transduction? Is there a lot of co-expression or just the opposite?

- as far as I know the seizures in the intrahippocampal kainic acid model are unilateral, so then I am really wondering whether the authors are really looking at seizure activity in the contralateral hippocampus.

- time profile of effects is compared between the BARNI and hM4Di approach. However you totally cannot compare it as the mechanisms are different. So this needs to be left out.

- the increase of acetylcholine during seizures is mentioned as an additional additive effect. Which could be right but anyway I don't see the benefit as the increase of ACh would be only during seizure, so to late to prevent seizures. The increase in response to seizures is also very short and given that we are looking at an ionotropic receptor system I would not influence the risk for a next seizure.

- The algorithm for seizure threshold testing is not fully clear:

10-400 μ A in 10-30 μ A steps?

- Do I understand well that it is possible that 3 spikes can be considered as a seizure according to the protocol: "Spike clusters greater than 3 seconds in duration with an inter-spike interval of less than 1 second were included as seizures."

- from figure 4E it seems that that there are also many cases where there is a drop of ACh during seizures?

Reviewer #2 (Remarks to the Author):

The authors report the development and use of a new chemogenetic tool, BARNI, Bradanicline- and Acetylcholine-activated Receptor for Neuronal Inhibition. BARNI is an engineered channel comprised of the $\alpha 7$ nicotinic acetylcholine receptor ligand-binding domain coupled to an $\alpha 1$ glycine receptor anion pore domain. BARNI can be activated by bradanicline, an $\alpha 7$ nicotinic acetylcholine receptor-selective agonist that has been used in human clinical trials in patients with schizophrenia. In this manuscript, the authors report that mice that have received hippocampal injections of the BARNI receptor expression cassette (vs. a 258 scrambled control cassette) via AAV-9 gene delivery with expression driven by a human synapsin promoter) demonstrate reduced hippocampal neuronal excitability, increased threshold for evoked seizures and a reduction in the frequency of spontaneous seizures following IHKA after treatment with bradanicline (to activate BARNI).

Overall, this is a well-done study with potential clinical significance. There is a substantial clinical need for new therapeutic approaches, such as chemogenetic approaches, for the 1/3 of patients with focal epilepsy whose seizures are not fully controlled with currently available antiseizure medications. As noted by the authors, BARNI has some significant potential advantages compared to other DREADDs that have been previously utilized in epilepsy models, including that bradanicline has been found safe in phase II clinical trials and that BARNI has some endogenous activity driven by acetylcholine, which the authors convincingly show is increased after seizures. It is difficult to assess the potential clinical utility, however, based only on the data as presented, and several clarifications and some additional data would significantly strengthen the manuscript.

It is noted on page 5 that “mice (are) virally transfected with vectors containing a pan-neuronal hSyn-BARNI expression cassette or a control scrambled cassette into the hippocampus”. Why was a pan-neuronal promoter such as synapsin chosen instead of an excitatory neuronal promoter (such as vGlut promoter or the modified CamKIIa promoter that is excitatory neuron specific)? Wouldn't there be a benefit to targeting inhibition only to excitatory neurons? Might that be more effective? Also, it is noted that “Neurons in the CA1 subregion were selected for recording”. Were these only CA1 pyramidal (excitatory) neurons? Since BARNI should have been transfected into inhibitory as well as excitatory neurons, recordings from both types of neurons would be informative.

A critical unanswered question is what is the effect on cognition of the enhanced inhibition resulting from BARNI at baseline (with endogenous Ach activation) and after activation with bradanicline? Mice were recorded during active running tasks which should have allowed for at least simple spatial navigation and memory performance to be assessed. Establishing if there is any effect of BARNI activation on cognition will be key to determining if this chemogenetic approach has translational potential.

There are several issues related to the EEG recording and seizure quantification that need to be clarified:

1. Page 18, it is stated that “mice were connected to a 24/7 video-EEG monitoring system to record the occurrence of seizures”, yet the data on seizures that is presented in Figure 3 (and Suppl Fig 5) only cover 6 hrs of recording (1 hr before and 5 hrs after bradanicline or vehicle injection). Supplemental Fig 7 shows cumulative number of seizures over 3.5 weeks, but does not include bradanicline-treated mice and it is unclear if this is 24/7 recording or a series of daily 6 hr recordings. Further, in Suppl Fig 7, the Y-axis is listed as “Normalized cumulative number of seizures”, but it is not stated what this is normalized to. The figure legend notes that “BARNI-expressing mice show a lower cumulative number of seizures at the end of the baseline recording period” (it looks like only after 20 days of recording). I am assuming this is because it takes ~3 weeks to have substantial transgene (BARNI cassette) expression from AAV. It would be worth seeing if the curves for the mice that received BARNI vs. scrambled cassette continue to differentiate over time, and also to compare longer term seizure frequency between scrambled cassette (control), BARNI-vehicle and BARNI-bradanicline treated mice over several weeks after day 21. Based on the PK data presented in Suppl Fig 4, repeated oral dosing of bradanicline should be feasible as at 4 hrs post-oral dose the mice are showing levels that are >10-fold higher than the BARNI EC50. Longer-term recordings will be critical for determining if the effects of BARNI-activation are sustained and if this strategy would be effective for long-term seizure suppression in people with epilepsy.
2. In Figure 3 and Suppl Figure 5 (bottom) it appears that seizure duration trends upward from 90-300 minutes after bradanicline injection. Was this intragroup effect over time significant? What happens after 300 minutes?
3. In Suppl Figure 5, absolute quantifications of seizure frequency are reported. There appears to be a relatively small difference between BARNI+Veh vs BARNI+Br, which is only statistically significant at some timepoints, and the BARNI+Br mice are still having 10-20 seizures per 30-minute period. The difference between Scrambled controls (+Veh or +Br) and BARNI+Br is greater (~2-fold and lower p-value), but it is unclear if it would be sustained since recording is only reported out to 5 hrs after Br or vehicle injection. As mentioned above, longer term EEG recordings with repeated twice daily dosing of bradanicline and reporting of both cumulative seizure number (as was done in Suppl Fig 7 for baseline in BARNI-expressing and Scramble-expressing mice) and seizures/unit time (per hour, day or week) would be helpful in assessing potential clinical utility.
4. In Figure 4, the authors provide compelling evidence for Ach increases in association with electrically evoked seizures. This was done while the mice were running on a treadmill. What is the reason for this?

Was there an associated behavioral assessment either to document if there was a behavioral correlate of the seizures or for any type of cognitive assessment?

5. On page 18, line 422-23, it is stated that for spontaneous seizures in IHKA treated mice, “Spike clusters greater than 3 seconds in duration with an inter-spike interval of less than 1 second were included as seizures”. This is substantially shorter than what most authors require to classify an event as a seizure, which is more typically 6-10 seconds. Why was this cut-off chosen? It is also different than the criteria used for evoked seizures (page 16, line 367-369), which is 6 seconds: “Evoked seizures were considered sustained periods (≥ 6 sec) of elevated amplitude hippocampal signal, characterized by rhythmic bursting at 2-4 Hz along with a higher frequency component (20-50 Hz), which began acutely (< 10 sec) after electrical stimulation ceased.” If the cut-off for spontaneous electrographic seizures is changed to 6 seconds, how does this impact the data?

6. For both spontaneous and evoked seizures, it would be helpful to know if there was a behavioral component to any of the seizures, and if so, what percentage of the electrographic seizures have a behavioral correlate? Does the number or percent of behavioral seizures differ between BARNI+Br vs. other groups?

Minor comment:

Introduction, paragraph 1, it is stated that “About one-third of epilepsy patients will not respond positively to anti-seizure medications”. The term “respond positively” is vague, and this sentence should be re-worded. The correct statistic is that one-third of patients will not have their seizures fully controlled by antiseizure medications.

Reviewer #3 (Remarks to the Author):

The paper by Nguyen et al deals with the development of a chemogenetic system to treat epilepsy. I am not expert in these fields, so I cannot comment about the experiments, but I have found some issues that need to be addressed.

The authors have built chimeric receptor made by the LBD of $\alpha 7$ nAChR and the pore domain of the $\alpha 1$ GlyR, whose activation reduces neuronal excitability. Such kind of eLGIC has been already described (ref 25, 39 and also in doi.org/10.7554/eLife.64241) but I could not understand if it the same chimera or a different one. Diversely to the other reports, the authors did not activate the chimera with a specifically-designed agonist but instead used TC5619. This strategy has a clear advantage: TC5619 is highly selective for $\alpha 7$ nicotinic receptors and it has been already tested in clinical trials, proving to be well tolerated without severe adverse effects. However, TC5619 is able to activate also the native $\alpha 7$ receptors, whose consequences are not easily predictable. The important beneficial effects of $\alpha 7$ activation on cognition, neuroprotection, and on the cholinergic anti-inflammatory pathway have been extensively described, unfortunately it also increases intracellular calcium concentration, glutamate

release, angiogenesis, cell proliferation and migration. Contrary to the statement reported at line 194, no clinically-approved selective alpha7 nAChR agonist is available. The only clinically approved nicotinic drug is varenicline which however modulates also heteromeric neuronal nicotinic receptors. Alpha7-selective agonists or partial agonists have been tested in several clinical trials for schizophrenia or to contrast the cognitive decline in Alzheimer's disease but none of them have been approved due to a lack of efficacy.

TC5619 may have another limitation: it is considered to be "modestly CNS penetrant" (see discussion in Canning et al [dx.doi.org/10.1124/jpet.121.000641](https://doi.org/10.1124/jpet.121.000641)). Indeed, figure S4 shows that the brain/plasma ratio is about 0.1. The authors have used a high TC5619 concentration for their studies (100 mg/kg); with such dose the range of concentrations measured in brain within 4 h are well above the EC50. Extrapolation of the concentration values from figure S4 shows that after 4 hours the brain concentration of TC5619 is still enough for a receptor activation of about 90%. I wonder if such high dose or high receptor activation are necessary for the antiepileptic activity; for their behavioral tests Hauser et al (ref40) administered a concentration about 2 orders of magnitude lower. The authors suggest that the strength of the effect is related to channel expression; which level of channel expression is needed? Probably this affects also the required dose.

The legend of Fig. S4 reports "Dotted lines in each plot indicate the EC50 for BARN1 channel activation recorded in dissociated hippocampal neurons": I think that the EC50 does not vary with time and I am expecting to see a straight line.

Please add somewhere in the paper the structure of TC5619 or at least its chemical name.

Responses to Referees

Reviewer #1:

With much interest I have read this paper. The authors developed an alternative PSAM receptor (but strangely the names PSAM-PSEM are nowhere mentioned in the paper) and showed suppression of seizures in the intrahippocampal kainic acid mouse model.

Author Response: Done. We thank the reviewer for their interest in our paper. Regarding the reviewer's comment, although we had referenced the PSAM engineered receptors in the introduction of our paper, our BARNI receptor itself is not the same as the PSAM receptor. To demonstrate this point we have assembled a new Supplementary Fig. 1, which contains a complete explanation of the differences between the BARNI receptor and other similar eLGICs, including the PSAMs. While both the BARNI receptor and PSAMs fall within the general category of eLGICs, our understanding is that the PSAMs are a specific subset of receptors where mutations have been specifically screened to identify selective responses to certain ligands. We agree with the suggestion of the Reviewer and have now included references by name to the related PSAMs in our introduction (page 4, lines 63-65) and results (page 5, lines 82-86). We believe that this will help clarify any confusion between our BARNI receptor and the PSAMs.

I have a couple of comments:

- Now the authors performed a single administration of the drug but what about repeated treatment?

Author Response: Done. We thank the reviewer for inquiring about repeated dosing, a topic which was also brought up by the other reviewers. To address this, we performed experiments where we administered a second i.p. injection of the same compound (either vehicle or bradanicline) 2 hours after the first injection. We selected the 2-hour timing based on our bradanicline concentration data in Supplementary Fig. 8. As shown by the new Supplementary Fig. 9, we were able to extend the time window of the seizure frequency reduction to more than twice as long compared to a single dose administration (Fig. 3). We are excited about these results, which bode well for the potential future clinical efficacy of our chemogenetic platform for epilepsy control.

- very few microscopy images are shown, especially regarding the expression of the acetylcholine biosensor and the calcium indicator. I think an illustrative image should be shown. The authors did perform 2-photon imaging. I am a bit worried about the fact that the serotypes for expression of the Ach and calcium sensor was the same. Is there no competition for transduction? Is there a lot of co-expression or just the opposite?

Author Response: Done. We have added a representative image of the calcium and ACh sensor expression to Figure 4. As noted in the methods, we found that the optimal ratio of virus to enable co-expression of both vectors in the majority of neurons was a 1:4 mixture of iAChSnFR to jRGECO1a virus. We observed no issues with the matching serotypes of our viruses negatively impacting the ability of neurons to co-express both sensors.

- as far as I know the seizures in the intrahippocampal kainic acid model are unilateral, so then I am really wondering whether the authors are really looking at seizure activity in the contralateral hippocampus.

Author Response: Done. We thank the reviewer for providing us the opportunity to make this important point: While seizures in the intrahippocampal kainic acid (IHKA) mouse model of temporal lobe epilepsy emerge from a unilateral focus, they often spread to the contralateral side. Previous work utilizing EEG recordings from subdural electrodes in IHKA mice has shown that seizure activity can be recorded above both hippocampal hemispheres (Lisgaras & Scharfman, *Neurobio. Disease* 2022). In addition, our previous work shows that closed-loop optogenetic excitation of parvalbumin interneurons

in either the contralateral or ipsilateral hemisphere significantly shortens seizure durations in IHKA mice (Krook-Magnuson, Nat. Comm 2013). As part of a separate study, we utilized multichannel silicon probe recordings locally within the ipsilateral and contralateral hemisphere of IHKA mice. We provide the below Reviewer Figure 1 to show that we are able to record epileptic activity from both hemispheres, with greater amplitudes in the ipsilateral hemisphere.

We have added an additional statement in our manuscript to note this point in page 9, lines 191-192: “Previous work has shown that seizures in the IHKA mouse model can spread and be detected on the contralateral side⁵¹.”

Reviewer Figure 1: Bilateral activation during spontaneous electrographic seizure in IHK model

A) Spike-triggered average local field potential (LFP) responses aligned to the peak of ipsilaterally detected interictal spikes. Channels are arranged by depth (50 μ m spacing) with the CA1 pyramidal cell layer and dentate granule cell layer denoted. Note the large amplitude response detected contralaterally with positive voltages detected ventral to the CA1 pyramidal cell layer. **B-C)**, raw LFP potential traces during non-convulsive electrographic seizure activity shows clear seizure patterns detected in the contralateral stratum radiatum (the channel displayed is 100 μ m below the pyramidal cell layer). C is an inset of activity in B.

- time profile of effects is compared between the BARNI and hM4Di approach. However you totally cannot compare it as the mechanisms are different. So this needs to be left out.

Author Response: Done. We agree with the reviewer that since the mechanisms underlying the seizure suppression using BARNI and hM4Di approaches are different we should avoid making detailed comparisons. While this is important to note, we also believe it is valuable to point out how two distinct strategies demonstrate the general promise of chemogenetics in suppressing seizures. We have now clarified the sentence at page 9, lines 212-215 to the following to address this: “Notably, DREADD seizure suppression was found to last for several hours, even after a single administration of CNO, and despite the difference underlying the mechanism of the effect, we also observed prolonged seizure suppression with BARNI after i.p. injection of bradanicline.”

- the increase of acetylcholine during seizures is mentioned as an additional additive effect. Which could be right but anyway I don't see the benefit as the increase of ACh would be only during seizure, so too late to prevent seizures. The increase in response to seizures is also very short and given that we are looking at an ionotropic receptor system it would not influence the risk for a next seizure.

Author Response: We thank the reviewer for bringing up this important topic. We would like to first point out that our data in Supplementary Fig. 11 indicates that solely expressing the ACh-sensitive BARN1 construct, in the absence of any bradanicline administration, is sufficient to reduce seizure occurrence. In addition, the duration of seizure intervention does not need to be long lasting, or fully aligned with seizure onset, in order to have a significant effect on seizures. Our previous work has shown that closed-loop optogenetic modulation of targeted cell types initiated 5 seconds after the detected start of a seizure is able to significantly shorten seizure duration (Krook-Magnuson et al., Nat. Comm. 2013), while also restoring normal brain function (Kim et al., Epilepsia 2020). Thus, on-demand approaches to control neuronal excitability has been shown to be effective at achieving seizure control.

- The algorithm for seizure threshold testing is not fully clear: 10-400 μA in 10-30 μA steps?

Author Response: Done. As suggested, we have now clarified in our methods that the step sizes were 10-100 μA : 10 μA steps; 100-240 μA : 20 μA steps, >240 μA : 30 μA steps.

- Do I understand well that it is possible that 3 spikes can be considered as a seizure according to the protocol: "Spike clusters greater than 3 seconds in duration with an inter-spike interval of less than 1 second were included as seizures."

Author Response: Done. The criteria we used to initially classify spike clusters aimed to be as inclusive as possible, which was then followed by experimenter verification of detected events. While theoretically 3 spikes could be detected as a potential seizure, this is rarely if ever the case and such events are then excluded upon subsequent review. We found that short seizures tended to have a higher frequency of spikes while longer seizures were comprised of spikes with more variable frequency. As shown in the below Reviewer Figure 2, longer seizures often had a ramping up and down of spiking frequency with the beginning and end of the seizure comprised of spikes with an inter-spike interval of around 1 second. Thus, this parameter was chosen in order to have as accurate an assessment of seizure duration as possible.

Reviewer Figure 2: Spike clustering

Representative seizure with insets showing spike frequency at the beginning (green) and towards the end (purple) of the seizure.

- from figure 4E it seems that that there are also many cases where there is a drop of ACh during seizures?

Author Response: Done. We thank the reviewer for pointing this out. Indeed, there are several seizures that were recorded where we saw a decrease in ACh signal during seizures, and these tended to be associated with lower (or negative) changes in the calcium signal. Previous work has shown that ACh decreases during sharp wave ripple oscillations (Zhang et al., PNAS 2021), and it is thought that the high frequency oscillations observed at the seizure focus in epilepsy are a pathological variant of sharp wave ripples. Since in our experiments we did not image the ipsilateral side of the brain, we cannot determine whether the drop in ACh signal is across both hemispheres, but this topic would be an intriguing area for future investigation. As we recorded electrical activity in the ipsilateral hippocampus, seizures not associated with combined increases in Ca^{2+} and ACh signals may represent events that failed to spread bilaterally to the contralateral hippocampus being imaged.

Reviewer #2:

The authors report the development and use of a new chemogenetic tool, BARNI, Bradanicline- and Acetylcholine-activated Receptor for Neuronal Inhibition. BARNI is an engineered channel comprised of the $\alpha 7$ nicotinic acetylcholine receptor ligand-binding domain coupled to an $\alpha 1$ glycine receptor anion pore domain. BARNI can be activated by bradanicline, an $\alpha 7$ nicotinic acetylcholine receptor-selective agonist that has been used in human clinical trials in patients with schizophrenia. In this manuscript, the authors report that mice that have received hippocampal injections of the BARNI receptor expression cassette (vs. a 258 scrambled control cassette) via AAV-9 gene delivery with expression driven by a human synapsin promoter) demonstrate reduced hippocampal neuronal excitability, increased threshold for evoked seizures and a reduction in the frequency of spontaneous seizures following IHKA after treatment with bradanicline (to activate BARNI).

Overall, this is a well-done study with potential clinical significance. There is a substantial clinical need for new therapeutic approaches, such as chemogenetic approaches, for the 1/3 of patients with focal epilepsy whose seizures are not fully controlled with currently available antiseizure medications. As noted by the authors, BARNI has some significant potential advantages compared to other DREADDs that have been previously utilized in epilepsy models, including that bradanicline has been found safe in phase II clinical trials and that BARNI has some endogenous activity driven by acetylcholine, which the authors convincingly show is increased after seizures. It is difficult to assess the potential clinical utility, however, based only on the data as presented, and several clarifications and some additional data would significantly strengthen the manuscript.

Author Response: Done. We thank the reviewer for their positive assessment of our manuscript. The following comments we believe have helped significantly strengthen our manuscript.

It is noted on page 5 that “mice (are) virally transfected with vectors containing a pan-neuronal hSyn-BARNI expression cassette or a control scrambled cassette into the hippocampus”. Why was a pan-neuronal promoter such as synapsin chosen instead of an excitatory neuronal promoter (such as vGlut promoter or the modified CamKIIa promoter that is excitatory neuron specific)? Wouldn't there be a benefit to targeting inhibition only to excitatory neurons? Might that be more effective?

Author Response: Done. We concur with the reviewer that there may be further benefit from more selective targeting of our BARNI receptor to particular cell types, rather than using a pan-neuronal promoter. Indeed, we agree that targeting BARNI to only excitatory neurons might provide even more effective seizure control and this is definitely a future direction to further improve on our chemogenetic platform. We have added this point to our discussion (page 10, lines 224-227):

“Interneurons, such as parvalbumin-expressing basket cells, are important drivers of memory-associated neuronal oscillations⁶²⁻⁶⁴. A potential avenue for further development of our chemogenetic platform is to target BARNI expression solely to excitatory neurons, which might provide even more effective seizure control while minimizing the potential for any cognitive impacts.”

Also, it is noted that “Neurons in the CA1 subregion were selected for recording”. Were these only CA1 pyramidal (excitatory) neurons? Since BARNI should have been transfected into inhibitory as well as excitatory neurons, recordings from both types of neurons would be informative.

Author Response: Done. While we did not recover recorded neurons for post-hoc confirmation of cell type, we did collect data on neuron intrinsic firing properties that allows us to differentiate putative fast-spiking interneurons vs putative pyramidal cells. Separating out neurons with action potential widths of 1 ms or less as likely interneurons, we see that BARNI channel activation decreases neuronal excitability in both populations (see new Supplementary Fig. 4). Given that the hSyn promoter should lead to pan-neuronal BARNI expression and that the Cl⁻ currents evoked by BARNI activation will have an inhibitory impact on the majority of adult neurons, our results are in line with expectations. We now mention these findings in the results section (page 5-6, lines 107-109), along with adding the relevant methods.

A critical unanswered question is what is the effect on cognition of the enhanced inhibition resulting from BARNI at baseline (with endogenous Ach activation) and after activation with bradanicline? Mice were recorded during active running tasks which should have allowed for at least simple spatial navigation and memory performance to be assessed. Establishing if there is any effect of BARNI activation on cognition will be key to determining if this chemogenetic approach has translational potential.

Author Response: Done. For analysis of spatial encoding memory tasks, mice need to have run a minimum number of laps on a belt with tactile and/or visual cues in order to record enough calcium and spatial information for proper decoding. To enable this, mice need to be trained over several weeks to run on the treadmill. Unfortunately, our mice were imaged on an uncued belt and not trained to run, and therefore rarely ran the minimum number of laps required for us to adequately assess their spatial encoding. We have added the clarification of an uncued belt to the methods. We would further note that none of our *in vivo* imaging experiments were performed in BARNI-expressing animals, as the GFP tag on the receptor would have interfered with the green iAChSnFr sensor.

To address the reviewer’s concerns, we have now performed hippocampal local field potential (LFP) recordings in BARNI and Scramble mice freely moving within an open field arena, following saline and bradanicline injections, providing initial insights into the cognition-related impacts of our approach (Supplementary Figure 7). Theta (4-10 Hz) and Gamma (20-90 Hz) hippocampal oscillations during active behavior are associated with memory processing, while ripples (90-200 Hz) within inactive periods support memory consolidation (Buzsaki, Hippocampus 2015). As noted in our discussion (page 10), bradanicline has been evaluated through multiple phase 2 clinical trials and been shown to be safe and well tolerated. Indeed, early experiments indicated bradanicline alone improves memory performance (Hauser et al., Biochemical Pharmacology 2009) and while these benefits were not affirmed in human trials, there were likewise no indications of negative cognitive impacts (Lieberman et al., Neuropsychopharmacology 2013; Walling et al., Schizophrenia Bulletin 2016). In agreement with those studies, we did not observe any substantial impacts of bradanicline administration with or without BARNI expression on these cognitively relevant oscillations, although our findings do not preclude the potential for more subtle effects on cognition and behavior. While our metrics show no statistically significant changes, impaired PV interneuron activity can disrupt ripples (Karlócai et al., Brain 2014; Schlingloff et al., J Neurosci. 2014; Gan et al., Neuron 2017), which could account for our raw data trending towards slightly reduced ripple occurrence and intra-ripple frequency in BARNI mice treated with bradanicline. Future targeting of BARNI specifically to pyramidal cells may further reduce those risks and lead to even better cognitive outcomes. Also, as BARNI receptor expression in the absence of bradanicline administration reduces seizure frequencies, while posing less risk of impacting ripple occurrence, further titration of drug doses may be sufficient to minimize any potential cognitive side effects of our approach. Lastly,

epileptiform activity itself produces substantial reductions in ripple occurrence (Karlócai et al., Brain 2014; Marchionni et al., Epilepsia Open 2019) and leads to disrupted memory, so the net impact on cognitive function with the BARNI approach, where seizures are ameliorated, will be further assessed in future studies. Indeed, prior studies suggest that suppression of seizures by itself might be sufficient to improve cognition (Helmstaedter et al., Seizure 2018; Wang et al., Brain and Behavior 2020; Popp et al., Epilepsy and Behavior 2021; Kim et al., Epilepsia 2020).

There are several issues related to the EEG recording and seizure quantification that need to be clarified: 1. Page 18, it is stated that “mice were connected to a 24/7 video-EEG monitoring system to record the occurrence of seizures”, yet the data on seizures that is presented in Figure 3 (and Suppl Fig 5 [now Fig. S8]) only cover 6 hrs of recording (1 hr before and 5 hrs after bradanicline or vehicle injection). Supplemental Fig 7 [now Fig. S11] shows cumulative number of seizures over 3.5 weeks, but does not include bradanicline-treated mice and it is unclear if this is 24/7 recording or a series of daily 6 hr recordings. Further, in Suppl Fig 7 [now Fig. S11], the Y-axis is listed as “Normalized cumulative number of seizures”, but it is not stated what this is normalized to.

Author Response: Done. We thank the reviewer for identifying these points for clarification. Indeed, the animals were recorded 24/7 over several weeks. Our seizure analysis graphs highlight the time course of the effect after drug administration, where seizure duration and frequency returns to baseline within five hours after i.p. injection of bradanicline. We selected not to include EEG recorded beyond the anticipated effective period of bradanicline in our presented analysis. Our cumulative seizure analysis was done on mice before bradanicline treatment, to evaluate whether even before drug administration we already see a significant effect of BARNI expression alone on seizure frequency. The cumulative number of seizures is normalized to the cumulative number of seizures observed at 14 days for each mouse, and this clarification has been added to the legend of Supplementary Figure 11.

The figure legend notes that “BARNI-expressing mice show a lower cumulative number of seizures at the end of the baseline recording period” (it looks like only after 20 days of recording). I am assuming this is because it takes ~3 weeks to have substantial transgene (BARNI cassette) expression from AAV. It would be worth seeing if the curves for the mice that received BARNI vs. scrambled cassette continue to differentiate over time, and also to compare longer term seizure frequency between scrambled cassette (control), BARNI-vehicle and BARNI-bradanicline treated mice over several weeks after day 21.

Author Response: Done. Yes, the reviewer is correct in that the differences in cumulative number of seizures between BARNI-expressing and control mice are only apparent after about 3 weeks due to the time required to have substantial transgene expression. The reviewer brings up an intriguing question regarding the long-term effects of our manipulations. We would like to clarify that our drug treatment experiments were done on the same mice used for the cumulative seizure plots after at least 21 days of recording, and that effects were averaged across at least two days of injection with either vehicle or bradanicline, as noted in the methods. This suggests that despite the effect of BARNI expression alone, bradanicline administration is still able to induce a significant reduction in seizure duration and frequency. These results support our suggestion that drug treatment can work in concert with ACh increases during seizures to provide an additive effect on BARNI receptor activation to control seizures.

Based on the PK data presented in Suppl Fig 4 [now Fig. S6], repeated oral dosing of bradanicline should be feasible as at 4 hrs post-oral dose the mice are showing levels that are >10-fold higher than the BARNI EC50. Longer-term recordings will be critical for determining if the effects of BARNI-activation are sustained and if this strategy would be effective for long-term seizure suppression in people with epilepsy.

Author Response: Done. We agree that oral bradanicline administration would be a good potential approach to maintain mice at doses above EC50 for long periods. However, for the current study we chose to use i.p. drug injections to remove

uncertainty over the precise drug concentrations mice were receiving, in a manner that was not dependent on their water consumption. We have now shown in our new Supplementary Figure 9 that a second dose administration 2 hours after the first is able to significantly prolong the reduction in seizure frequency to more than 10 hours. These new results support the idea that repeated dosing will be able to provide more sustained seizure control.

2. In Figure 3 and Suppl Figure 5 (bottom) [now Fig. S8] it appears that seizure duration trends upward from 90-300 minutes after bradanicline injection. Was this intragroup effect over time significant? What happens after 300 minutes?

Author Response: Done. The upward trend was not significant for either normalized or absolute seizure durations. We see this same trend in our new 2-dose data as well (Supplementary Fig. 9) but again we did not see a significant increase in seizure duration at any time point after injection.

3. In Suppl Figure 5 [now Fig. S8], absolute quantifications of seizure frequency are reported. There appears to be a relatively small difference between BARNI+Veh vs BARNI+Br, which is only statistically significant at some timepoints, and the BARNI+Br mice are still having 10-20 seizures per 30-minute period.

Author Response: Done. We chose to use the normalized data for our main figure but included the absolute quantifications for completeness. A major caveat of the absolute quantifications, however, is that they do not take into account the variability in seizure frequency and duration between animals. Thus, the analysis shows greater variability, which confounds the ability to extract significant results. In addition, we have included both high and low BARNI expressing animals in our datasets, as shown in Supplementary Figure 10, which further contributes to the variability in our data. The small difference between BARNI+Veh vs BARNI+Br is also in line with our finding that BARNI expression alone results in lower seizure frequency, which would reduce the extent of the effect after bradanicline administration.

The difference between Scrambled controls (+Veh or +Br) and BARNI+Br is greater (~2-fold and lower p-value), but it is unclear if it would be sustained since recording is only reported out to 5 hrs after Br or vehicle injection. As mentioned above, longer term EEG recordings with repeated twice daily dosing of bradanicline and reporting of both cumulative seizure number (as was done in Suppl Fig 7 [now Fig. S11] for baseline in BARNI-expressing and Scramble-expressing mice) and seizures/unit time (per hour, day or week) would be helpful in assessing potential clinical utility.

Author Response: Done. As noted above, we have now added a new Supplementary Figure 9 which shows that a second administration of bradanicline 2 hours after the first is able to significantly prolong the reduction in seizure frequency to more than 10 hours in BARNI expressing mice. These results are taken from averages across several days of injections for each mouse. We would like to point out that our experimental design was such that each mouse had days of bradanicline injection interleaved with days of saline injection in order for us to properly perform a pairwise comparison. We are excited to follow up with these findings in future studies to investigate the long term viability of our approach. As an initial test, we tracked the effect of 2-dose bradanicline injection in a BARNI expressing mouse over several weeks. As shown in the below Reviewer Figure 3, we observed a consistent magnitude of seizure reduction 5 hours after the first bradanicline injection across more than 3 weeks.

Reviewer Figure 3: Viability of BARNI activation on seizures over time

Normalized seizure frequency 5 hours after first bradanicline dose in a 2-dose drug regimen (second dose administered 2 hours after the first) across several days in a BARNI-expressing mouse.

4. In Figure 4, the authors provide compelling evidence for Ach increases in association with electrically evoked seizures. This was done while the mice were running on a treadmill. What is the reason for this? Was there an associated behavioral assessment either to document if there was a behavioral correlate of the seizures or for any type of cognitive assessment?

Author Response: Done. We thank the reviewer for identifying this point for clarification. As noted in the methods, our imaging was done on head-fixed mice allowed to freely run on a treadmill. This is the standard setup for our imaging experiments to allow mice the ability to run while their heads are stationary, in order to minimize stress that complete movement restriction would have on the animals and to avoid the alternative need to perform experiments in anesthetized mice. We did not observe any clear behavioral correlate for the elicited seizures, and as noted in our earlier response, we used an uncued belt and did not train the mice to run enough laps to be able to adequately assess spatial encoding in our animals.

5. On page 18, line 422-23, it is stated that for spontaneous seizures in IHKA treated mice, “Spike clusters greater than 3 seconds in duration with an inter-spike interval of less than 1 second were included as seizures”. This is substantially shorter than what most authors require to classify an event as a seizure, which is more typically 6-10 seconds. Why was this cut-off chosen? It is also different than the criteria used for evoked seizures (page 16, line 367-369), which is 6 seconds: “Evoked seizures were considered sustained periods (≥ 6 sec) of elevated amplitude hippocampal signal, characterized by rhythmic bursting at 2-4 Hz along with a higher frequency component (20-50 Hz), which began acutely (< 10 sec) after electrical stimulation ceased.” If the cut-off for spontaneous electrographic seizures is changed to 6 seconds, how does this impact the data?

Author Response: Done. Within the spectrum of human epilepsy syndromes, there is wide variability of typical seizure durations, ranging from < 1 sec to tens of minutes (Larsen et al., *Epilepsia* 2023; Zuberi et al., *Epilepsia* 2022; Specchio et al., *Epilepsia* 2022). Agreed upon seizure duration thresholds are unfortunately lacking across diverse rodent experimental models, where epileptiform events of < 1 sec may sometimes be considered in analysis (Löscher, *Epilepsy Res.* 2016). We believe we are justified in using different seizure criteria for the two different models for eliciting seizures. In general, our evoked seizures were longer than our spontaneous seizures. In addition, we observed a large variability in seizure duration in our IHKA mice, both within the same mouse and between different mice. In order to enable us to capture the full range of seizure durations, especially after bradanicline administration in BARNI expressing animals which leads to a slight reduction in seizure duration, we used a lower threshold to analyze our IHKA mice.

However, we believe the reviewer presents a great question of whether the seizure duration threshold we selected for the IHKA experiments impacted our finding. To address this, we have reanalyzed a subset of our IHKA data using a 6-second threshold and found the same results, with a significant reduction in seizure frequency, shown below in Reviewer Figure 4. We have now noted this in our methods (page 19, line 470-471): “Similar results were obtained with analysis using a longer 6-second seizure threshold.” We would like to point out that unlike our original analysis with a 3-second threshold where we observed a brief but significant reduction also in seizure duration, using a 6-second threshold led to no difference in seizure duration at any time point after drug administration. This supports our decision to use a shorter threshold in order to include more epileptic events.

Reviewer Figure 4: Seizure analysis using 6 second threshold

Left, frequency of spontaneous seizures decreased after a single i.p. injection of bradanicline. 30 minute bins. ($F(33, 198)=1.951, P=0.0028$, two-way RM ANOVA, Time x Vector + Drug). *Right*, duration of spontaneous seizures observed after a single i.p. injection of bradanicline. 30 minute bins, dashed line indicates time of second dose. ($F(33, 198)=1.809, P=0.0072$, two-way RM ANOVA, Time x Vector + Drug). Significance values on graph are shown for comparison between vehicle and bradanicline treatment in BARNI-expressing mice at each time point using paired t-test. Significance values in legend are shown for comparison between vector expression and treatment groups using Tukey’s multiple comparisons test. Mean \pm SEM, $n = 6$ Scramble, 5 BARNI mice. * $P<0.05$, ** $P<0.01$, **** $P<0.0001$.

6. For both spontaneous and evoked seizures, it would be helpful to know if there was a behavioral component to any of the seizures, and if so, what percentage of the electrographic seizures have a behavioral correlate? Does the number or percent of behavioral seizures differ between BARNI+Br vs. other groups?

Author Response: Done. The frequency of overt behavioral seizures (\geq Racine Stage 3) is very low compared to the frequency of electrographic seizures (<1 overt behavioral seizure per hour vs >50 electrographic seizures per hour) in our mice. We quantified the number of overt behavioral seizures observed during the 5 hour time window after either vehicle or bradanicline injection in a subset of Scramble or BARNI expressing mice and found no significant difference between our groups. However, the low frequency of behavioral seizures, variability in the number of behavioral seizures observed between mice and across multiple days, and the limited time window to observe an effect are significant confounding factors that constrain our ability to rigorously test for significant effects from our analysis. Future experiments employing more long-term dosing and recordings across multiple weeks to obtain enough samples of behavioral seizures will be able to provide a more accurate assessment of our manipulations on behavioral seizures.

While we did not specifically track behavioral responses associated with electrically evoked seizures, the incidence of behavioral seizures beyond an approximate Racine score of 3 (note that some behaviors like head nodding could not occur in head-fixed mice) was very rare (an estimated $< 5\%$ of overall trials). Only one seizure had an electrographic component lasting >40 sec (80.8 sec) out of 50 total trials, which also had a clearly expressed behavioral element, and this was in a BARNI+Veh trial.

Minor comment:

Introduction, paragraph 1, it is stated that “About one-third of epilepsy patients will not respond positively to anti-seizure medications”. The term “respond positively” is vague, and this sentence should be re-worded. The correct statistic is that one-third of patients will not have their seizures fully controlled by antiseizure medications.

Author Response: Done. We thank the reviewer for this suggestion and have revised the sentence in page 3, lines 33-34 to: “About one-third of epilepsy patients will not have their seizures fully controlled by anti-seizure medications...”. Our original wording was intended to also acknowledge the population of people with epilepsy who cease taking medications that may successfully control their seizures, yet produce unacceptably severe side effects. However, we agree our previous wording was unnecessarily vague.

Reviewer #3:

The paper by Nguyen et al deals with the development of a chemogenetic system to treat epilepsy. I am not expert in these fields, so I cannot comment about the experiments, but I have found some issues that need to be addressed.

The authors have built chimeric receptor made by the LBD of alpha7 nAChR and the pore domain of the alpha1 GlyR, whose activation reduces neuronal excitability. Such kind of eLGIC has been already described (ref 25, 39 and also in doi.org/10.7554/eLife.64241) but I could not understand if it the same chimera or a different one.

Author Response: Done. The full sequence for the original (Grutter et al., PNAS 2005) α 7/Gly chimeric receptor is not included in their publication, but appears to match the “unmodified” α 7-Gly chimeric receptor assessed in the PSAM studies (Magnus et al., Science 2011 and Magnus et al., Science 2019) and which also served as the basis for the BARNI receptor. Distinct from the PSAMs, our BARNI receptor integrates the α 1GlyR Cys loop substitution described in the Grutter α 7(Cys-L)/Gly variant, which was found to produce faster receptor gating kinetics. With a focus on increasing sensitivity to synthetic PSEM ligands over endogenous ligands, different PSAMs have included mutations at W77, Q79, L131, Q139, L141 and Y217 (Magnus et al., Science 2011 and Magnus et al., Science 2019), which were not included in either the Grutter α 7/Gly or BARNI receptors. Unique to the BARNI chimeric receptor, we have used the α 1Ins splice variant of the α 1GlyR, which includes an additional 8 amino acid insert in the M3-M4 loop portion of the ion pore domain (K336_E337insSPMLNLFQ). GlyR with the α 1Ins splice variant display larger peak currents than the α 1 Δ Ins variant (Raltshev et al., JBC 2016) used in the other eLGICs. We now mention these distinctions at the start of our results section (page 5, lines 82-86) and include a new Supplementary Fig. 1 showing the amino acid sequence of the BARNI channel, along with where it varies from the original (Grutter et al., PNAS 2005) α 7/Gly chimeric receptor.

Diversely to the other reports, the authors did not activate the chimera with a specifically-designed agonist but instead used TC5619. This strategy has a clear advantage: TC5619 is highly selective for alpha7 nicotinic receptors and it has been already tested in clinical trials, proving to be well tolerated without severe adverse effects. However, TC5619 is able to activate also the native alpha7 receptors, whose consequences are not easily predictable. The important beneficial effects of alpha7 activation on cognition, neuroprotection, and on the cholinergic anti-inflammatory pathway have been extensively described, unfortunately it also increases intracellular calcium concentration, glutamate release, angiogenesis, cell proliferation and migration.

Contrary to the statement reported at line 194, no clinically-approved selective alpha7 nAChR agonist is available. The only clinically approved nicotinic drug is varenicline which however modulates also heteromeric neuronal nicotinic receptors. Alpha7-selective agonists or partial agonists have been tested in several clinical trials for schizophrenia or to contrast the cognitive decline in Alzheimer’s disease but none of them have been approved due to a lack of efficacy.

Author Response: Done. We thank the reviewer for this clarification. We realize the sentence (new location page 9, line 205) was confounding the clinical approval status of $\alpha 7$ -selective agonists with nAChR modulators in general and have changed the word “approved” to “tested” to reflect this.

The reviewer is correct in pointing out that TC5619 (bradanicline) is also able to activate native $\alpha 7$ nAChRs, yet we found no measurable effect of bradanicline administration on hippocampal neuron excitability or normalized seizure dynamics in Scrambled control animals. In addition, we found no clear impact of bradanicline administration on hippocampal oscillations in control animals in new experiments we have now performed (Supplementary Fig. 7). Together, these results suggest that any activation of native $\alpha 7$ nAChRs by the dosage of bradanicline used in these experiments does not have a notable effect on hippocampal functions, including cognitive function. However, we fully concur that further studies of potential TC5619 side effects related to alterations in intracellular calcium concentration, glutamate release, angiogenesis, cell proliferation and migration should be conducted as part of future translational efforts.

TC5619 may have another limitation: it is considered to be “modestly CNS penetrant” (see discussion in Canning et al [dx.doi.org/10.1124/jpet.121.000641](https://doi.org/10.1124/jpet.121.000641)). Indeed, figure S4 [now Fig. S6] shows that the brain/plasma ratio is about 0.1. The authors have used a high TC5619 concentration for their studies (100 mg/kg); with such dose the range of concentrations measured in brain within 4 h are well above the EC50. Extrapolation of the concentration values from figure S4 shows that after 4 hours the brain concentration of TC5619 is still enough for a receptor activation of about 90%. I wonder if such high dose or high receptor activation are necessary for the antiepileptic activity; for their behavioral tests Hauser et al (ref40) administered a concentration about 2 orders of magnitude lower. The authors suggest that the strength of the effect is related to channel expression; which level of channel expression is needed? Probably this affects also the required dose.

Author Response: Done. We thank the reviewer for pointing out these important considerations. Indeed, the level of channel expression will be a critical determinant to the extent of effect seen with drug administration, as suggested by Supplemental Figure 10. Given the variability in channel expression observed in our cohorts, we chose a dose that would maximize activation of the receptors that are expressed. We agree that if given optimal expression across all targeted neurons in the region where activity needs to be suppressed, a lower dosage may be sufficient. In acute brain slices, we find that reducing the TC5619 dose from from EC95 (150 nM) to EC80 (50 nM) levels still suppresses neuronal excitability, although somewhat more modestly (Reviewer Fig. 5). A future direction will be to test if repeated injections at lower doses would be just as effective in controlling seizures. While we believe TC5619 is a good initial activator candidate, due to the relatively good safety profile observed in prior phase 2 clinical trials, further screening of activator compounds with improved physiochemical properties is a promising direction of research to pursue in the future.

Reviewer Figure 5. BARNI activation dose response

Time course of BARNI-expressing CA1 neuron responses to a 1 min bath application of varied bradanicline doses (150 or 50 nM). n = 7 cells/4 animals 50 nM, n = 18 cells/8 animals 150 nM.

The legend of Fig. S4 [now Fig. S6] reports “Dotted lines in each plot indicate the EC50 for BARNI channel activation recorded in dissociated hippocampal neurons”: I think that the EC50 does not vary with time and I am expecting to see a straight line.

Author Response: Done. We agree the prior inclusion of both dotted and dashed lines in Fig. S6 was confusing. We have now changed the color of the straight line at 6.87 ng/ml to more clearly illustrate the BARNI EC50.

Please add somewhere in the paper the structure of TC5619 or at least its chemical name.

Author Response: Done. We thank the reviewer for this suggestion. We have now added the structure of TC5619 to Figure 1 and its CAS registry number to the methods section.

REVIEWERS' COMMENTS

Reviewer #1 (Remarks to the Author):

I agree with the changes made and with the publication of this paper.

Reviewer #2 (Remarks to the Author):

The authors have addressed the majority of my concerns. I have one remaining recommendation. In their rebuttal, to address the concern regarding defining interictal epileptiform discharges as short as 3 seconds being classified as seizures, the authors presented a "reviewer figure" showing data using a classification of electrographic seizures with a 6-second threshold. This figure was very informative, but they did not include this figure in the revised manuscript. Instead, the authors added a sentence to the methods section stating that this analysis was done and yielded a similar result to that done using a 3 second criteria for electrographic seizures. In their rebuttal, the authors point out that in human epilepsy syndromes "there is wide variability of typical seizure durations", and "across rodent experimental models,...epileptiform events of <1 second may sometimes be considered in the analysis". This may be the case for generalized epilepsies, however the IHKA model used in the current study produces focal seizures of temporal lobe origin which are not typically that short either in rodents or humans. It would be more in line with accepted practice to classify events of <5-6 seconds that do not have a behavioral correlate (which these do not) as interictal epileptiform discharges and classify electrographic events of >5-6 seconds as seizures and provide figures with separate analysis of both of these data sets within the manuscript.

Reviewer #3 (Remarks to the Author):

The authors have addressed all my issues, and I do not have other remarks.

For me the paper can be accepted in the present form

Responses to Referees

Reviewer #2:

The authors have addressed the majority of my concerns. I have one remaining recommendation. In their rebuttal, to address the concern regarding defining interictal epileptiform discharges as short as 3 seconds being classified as seizures, the authors presented a “reviewer figure” showing data using a classification of electrographic seizures with a 6-second threshold. This figure was very informative, but they did not include this figure in the revised manuscript. Instead, the authors added a sentence to the methods section stating that this analysis was done and yielded a similar result to that done using a 3 second criteria for electrographic seizures. In their rebuttal, the authors point out that in human epilepsy syndromes “there is wide variability of typical seizure durations”, and “across rodent experimental models,...epileptiform events of <1 second may sometimes be considered in the analysis”. This may be the case for generalized epilepsies, however the IHKA model used in the current study produces focal seizures of temporal lobe origin which are not typically that short either in rodents or humans. It would be more in line with accepted practice to classify events of <5-6 seconds that do not have a behavioral correlate (which these do not) as interictal epileptiform discharges and classify electrographic events of >5-6 seconds as seizures and provide figures with separate analysis of both of these data sets within the manuscript.

Author Response: Done. We thank the reviewer for their careful review of our revised manuscript. We have followed their recommendation and included our seizure analysis using a 6-second threshold as Supplementary Figure 10. We have also revised the manuscript text (page 7, lines 151-154), pointing out that we obtained similar results using the different seizure inclusion criteria:

“Overall, we found a significant decrease in seizure frequency following administration of bradanicline in BARNI-expressing animals compared to either vehicle injection or Scramble-expressing animals, with these findings being consistent across both normalized and absolute seizure frequency quantifications and with the use of longer thresholds for seizure duration (Fig. 3c, Supplementary Fig. 9, 10).”